# INFOBATCH: LOSSLESS TRAINING SPEED UP BY UNBIASED DYNAMIC DATA PRUNING

**Ziheng Qin**[1][*] **Kai Wang**[1][*][†] **Zangwei Zheng**[1] **Jianyang Gu**[1] **Xiangyu Peng**[1] **Zhaopan Xu**[1]
**Daquan Zhou**[1] **Lei Shang**[2] **Baigui Sun**[2] **Xuansong Xie**[2] **Yang You**[1][‡]
[1]National University of Singapore    [2]Alibaba Group
{zihengq, kai.wang, youy}@comp.nus.edu.sg

## ABSTRACT

Data pruning aims to obtain lossless performances with less overall cost. A common approach is to filter out samples that make less contribution to the training. This could lead to gradient expectation bias compared to the original data. To solve this problem, we propose **InfoBatch**, a novel framework aiming to achieve lossless training acceleration by unbiased dynamic data pruning. Specifically, InfoBatch randomly prunes a portion of less informative samples based on the loss distribution and rescales the gradients of the remaining samples to approximate the original gradient. As a plug-and-play and architecture-agnostic framework, InfoBatch consistently obtains lossless training results on classification, semantic segmentation, vision pertaining, and instruction fine-tuning tasks. On CIFAR10/100, ImageNet-1K, and ADE20K, InfoBatch losslessly saves 40% overall cost. For pertaining MAE and diffusion model, InfoBatch can respectively save 24.8% and 27% cost. For LLaMA instruction fine-tuning, combining InfoBatch and the recent coreset selection method (DQ) can achieve 10 times acceleration. Our results encourage more exploration on the data efficiency aspect of large model training. Code is publicly available at NUS-HPC-AI-Lab/InfoBatch.

## 1 INTRODUCTION

In the past decade, deep learning has achieved remarkable progress in the computer vision area (Dosovitskiy et al., 2020; Touvron et al., 2020). Most state-of-the-art methods (Dosovitskiy et al., 2020; Touvron et al., 2020) are trained on ultra-large-scale datasets, but the heavy training cost is hardly affordable for researchers with limited computation resources. Reducing the training effort for large-scale datasets has become urgent for broader computer vision and other deep-learning applications.

An intuitive solution is to reduce the training sample amount. Dataset distillation (Nguyen et al., 2021; Zhao & Bilen, 2021; Wang et al., 2022) and coreset selection (Har-Peled & Mazumdar, 2004; Park et al., 2022; Xia et al., 2023) respectively synthesize or choose an informative dataset/subset from the original large one. Although the sample amount is reduced, the distillation and selection algorithms lead to extra costs. Besides, these two methods are also hard to achieve lossless performance.

The other solution is weighted sampling methods (Zhao & Zhang, 2014; Csiba & Richtárik, 2016; Johnson & Guestrin, 2018) that aim to improve the sampling frequency of certain samples. It improves the convergence speed, but their accelerations are sensitive to models and datasets (Zhao & Zhang, 2014). In another way, LARS (You et al., 2017) and LAMB (You et al., 2019) enable a super large batch size to improve data parallelism during training to reduce the overall training time. However, more computation units are required and the total training cost is not reduced, which limits the effect under constrained computation resources.

Most recently, a series of works propose to accelerate training by reducing the total training iterations. Toneva et al. (2018); Paul et al. (2021) estimate a score for each sample and accordingly prune less informative samples. Since those pruned samples are never added back during training, we refer

---

[*]Equal Contribution.

[†]Project Lead.

[‡]Corresponding author

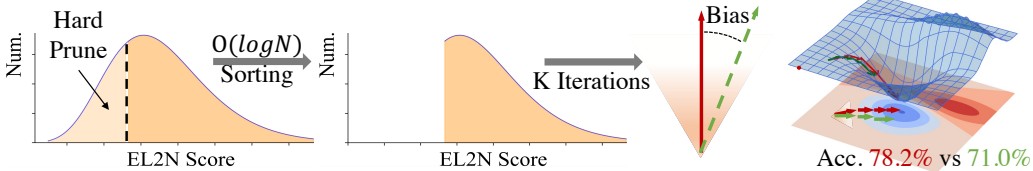

(a) Previous methods prune samples via setting some heuristic metrics (e.g. EL2N score). The hard pruning operation results in biased gradient expectation (green lines) and sub-optimal training results.

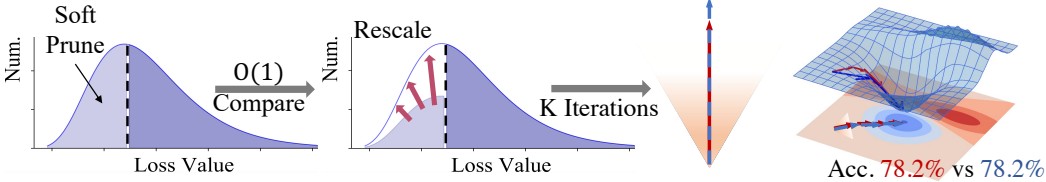

(b) InfoBatch soft prunes samples with small loss values and rescales the updates. Thereby InfoBatch maintains approximately the same gradient expectation direction (blue lines) as training on the original dataset.

Figure 1: Illustration of difference between InfoBatch and EL2N (Paul et al., 2021)(a static pruning method). InfoBatch achieves lossless acceleration performance, while EL2N has a performance drop of 7.2%. Experiments are conducted with ResNet-18 under the same pruning ratio of 50% on CIFAR-100. The semi-transparent triangles represent the variance range of gradient estimation.

to these methods as static pruning methods. However, these methods usually need several trials to estimate more accurate scores, which requires extra $O(MNT)$ overhead ($M$ is the model size, $N$ is the dataset size, and $T$ is the trial number), sometimes even longer than the training time on large-scale datasets (*e.g.* ImageNet-1K as in Tab. 2). Therefore, it is cumbersome to apply those static pruning methods (Toneva et al., 2018; Paul et al., 2021) on large-scale datasets.

To reduce this heavy overhead, Raju et al. (2021) dynamically prunes the samples based on easily attainable scores, e.g., loss values, during the training without trials. This work sorts the whole dataset by the score in each pruning cycle, which leads to $O(logN)$ per sample time complexity on sorting. It would cost much less on ImageNet-1k than static pruning, while the overhead would still be non-negligible on larger datasets. Meanwhile, directly pruning data may lead to a biased gradient estimation as illustrated in Fig. 1a, which affects the convergence result. This is a crucial factor that limits their performance, especially under a high pruning ratio (corresponding results in Tab. 1).

To tackle these issues, we propose InfoBatch, a novel unbiased dynamic data pruning framework based on the idea of maintaining the same expected total update between training on the pruned and original datasets. We refer to this idea as expectation rescaling for simplicity. Specifically, given a dataset, we maintain a score of each sample with its loss value during forward propagation. We randomly prune a certain portion of small-score (*i.e.* well-learned) samples in each epoch. Different from previous methods where well-learned samples are dropped directly, as shown in Fig. 1b, we scale up the gradient of those remaining small-score samples to keep an approximately same gradient expectation as the original dataset.

Compared to previous works (Toneva et al., 2018; Paul et al., 2021; Raju et al., 2021), the gradient expectation bias in optimization between InfoBatch and standard training is reduced, as illustrated in Fig. 1a and 1b. To further improve the performance stability and reduce the variance during convergence, full dataset is used for training in the last few epochs. Detailed analysis is provided in Sec 2.3 and Appendix B to prove the unbiasedness of InfoBatch.

InfoBatch is compatible with various deep-learning tasks. In this work, we investigate its effect on classification, semantic segmentation, vision pertaining, and language model instruction finetuning. With the same hyperparameters in most cases, InfoBatch achieves lossless training performances with $20\% \sim 40\%$ less overall cost across various tasks and architectures. It helps mitigate the heavy computation cost for training on ultra-large-scale datasets. The time complexity of InfoBatch is $O(1)$

per sample, which further accelerates from previous $O(logN)$ dynamic pruning methods, costing only seconds to reduce hours of training on datasets like ImageNet-1K.

## 2 HOW WE CAN ACHIEVE LOSSLESS RESULT

### 2.1 PRELIMINARIES

**Static Pruning.** Given a dataset $\mathcal{D} = \{z_i\}|_{i=1}^{|\mathcal{D}|} = \{(x_i, y_i)\}|_{i=1}^{|\mathcal{D}|}$, a score $\mathcal{H}(z)$ can be defined for each sample. For the pruning process, samples are discarded by a pruning probability $\mathcal{P}$ defined on top of $\mathcal{H}$. Static pruning directly discards all samples satisfying a certain condition before training, resulting in $\mathcal{P}(z; \mathcal{H}) \in \{0, 1\}$. For examples, Toneva et al. (2018) defines:

$$\mathcal{P}(z; \mathcal{H}) = \mathbb{1}(\mathcal{H}(z) < \bar{\mathcal{H}}), \tag{1}$$

where $\bar{\mathcal{H}}$ is a threshold and $\mathbb{1}(\cdot)$ is indicator function. A subset $\mathcal{S}$ is formed by pruning samples with $\mathcal{P}(z; \mathcal{H}) = 1$. Fig. 1a briefly illustrates the whole process of static pruning. A more detailed theoretical analysis is provided in Appendix B.

**Dynamic Pruning.** For dynamic pruning, pruning is done across training, and the score $\mathcal{H}_t$ can change along training steps, where $t$ denotes the temporal status. The probability is step-dependent:

$$\mathcal{P}_t(z) = \mathcal{P}(z; \mathcal{H}_t), \tag{2}$$

and forms a dynamically pruned dataset $\mathcal{S}_t$. Compared to static pruning, dynamic pruning has access to all the original data during training. Thus the gradient expectation bias should be much smaller than static pruning. However, such a scheme still has the following limitations: i). As claimed in Raju et al. (2021), low-score samples of different $t$ during the training could easily overlap. Directly pruning them every time may still lead to a bias (see in Fig. 1a). ii). Pruning samples leads to a reduced number of gradient updates. Under the premise of saving training cost, existing dynamic pruning methods Raju et al. (2021) hardly achieve lossless results compared to training on the original dataset. iii). Scoring and sorting operations in dynamic pruning are conducted repeatedly, the overhead of which is still non-negligible on large-scale datasets.

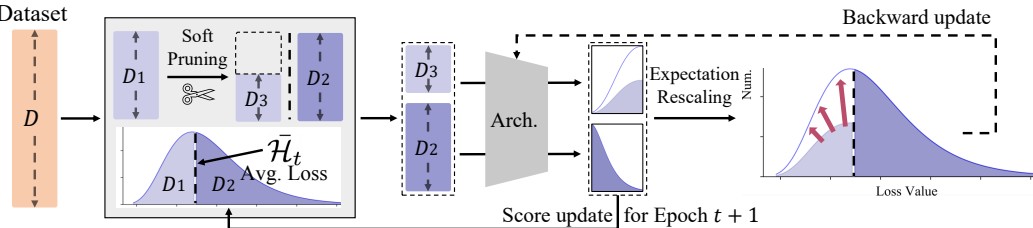

Figure 2: Illustration of the proposed *InfoBatch* framework. InfoBatch mainly consists of two operations, named soft pruning and expectation rescaling. $\bar{\mathcal{H}}_t$ denotes the adaptive thresholds of scores of samples. Soft pruning randomly prunes some samples from $\mathcal{D}_1$ with relatively small scores. For remaining samples from $\mathcal{D}_1$, expectation rescaling scales up the losses to keep the approximately same gradient expectation as the original dataset.

### 2.2 OVERVIEW OF INFOBATCH

Based on the above observation and analysis, we propose InfoBatch, a novel framework for achieving lossless training acceleration based on unbiased dynamic data pruning. As illustrated in Fig. 2, a score is maintained for each sample with its loss value during forward propagation. The mean of these values are set as the pruning threshold. A certain portion of small-score samples is accordingly pruned in each epoch. Then, to obtain the same expectation of gradient as the original dataset in each epoch, the gradients of the remaining small-score samples are scaled up. By doing this, compared to previous static and dynamic pruning methods, InfoBatch mitigates the performance differences between training on the pruned dataset and the original dataset. In order to further reduce the remaining gradient expectation bias, we train with the full dataset in the last few epochs.

## 2.3 UNBIASED PRUNE AND RESCALE

InfoBatch adopts the dynamic pruning process as in Eqn. 2. We first define its pruning policy $\mathcal{P}_t$. Previous methods using a deterministic pruning operation could cause bias. In contrast, InfoBatch introduces randomness into the pruning process. Given a dataset $\mathcal{D}$, in $t$-th epoch, a pruning probability is assigned to each sample based on its score. Such a soft pruning policy is formulated as:

$$\mathcal{P}_t(z) = \begin{cases} r, & \mathcal{H}_t(z) < \bar{\mathcal{H}}_t \\ 0, & \mathcal{H}_t(z) \geq \bar{\mathcal{H}}_t \end{cases}, \tag{3}$$

where $\bar{\mathcal{H}}_t$ is the mean value of all the scores $\mathcal{H}_t$ and $r \in (0, 1)$ is a predefined hyper-parameter as the pruning probability. This new prune policy has the following benefits: i). Soft pruning allows each small-score sample to be utilized for training, which reduces the bias caused by hard pruning in previous dynamic pruning methods. ii). The proposed strategy is based on the comparison with $\bar{\mathcal{H}}_t$, with no requirement to sort the whole training samples, which reduces the time complexity from $O(logN)$ to $O(1)$. It indicates that InfoBatch could be more efficient on large-scale datasets.

Then, we utilize loss values $\mathcal{L}(z)$ of each sample as the corresponding score based on the following two reasons: i). loss values can be obtained without extra cost, ii). loss values reflect the learning status of samples (Cilimkovic, 2015). Specifically, before the $t$-th ($t > 0$) epoch, we utilize the soft pruning policy to prune samples based on their scores. Then for the pruned samples, their scores remain unmodified as previous. For the remaining samples, their scores are updated by the losses in the current epoch. Mathematically, $\mathcal{H}_t(z)$ would be updated by the latest losses to $\mathcal{H}_{t+1}(z)$ by:

$$\mathcal{H}_{t+1}(z) = \begin{cases} \mathcal{H}_t(z), & z \in \mathcal{D} \backslash \mathcal{S}_t \\ \mathcal{L}(z), & z \in \mathcal{S}_t \end{cases}. \tag{4}$$

Note that, for the first epoch, we initialize the scores with $\{1\}$ provided no previous loss.

There are several benefits of the soft pruning policy, yet it still cannot avoid the influence caused by the reduced number of gradient updates. To address this issue, we scale up the gradients of the remaining samples. Specifically, given a remaining sample with score $\mathcal{H}(z) < \bar{\mathcal{H}}_t$, whose corresponding pruning probability is $r$, its gradient is scaled to $1/(1-r)$ times of original. For the samples with scores no less than $\bar{\mathcal{H}}_t$, the loss is not modified. Thereby the gradient update expectation is approximately equal to training on the original dataset. Besides, as the rescaling is operated on small-score samples, it further refines the direction of gradient update expectation. We provide the following theoretical analysis to demonstrate the necessity and advantages of expectation rescaling.

**Theoretical Analysis.** We can interpret the training objective as minimizing empirical risk $\mathcal{L}$. Assuming all samples $z$ from $\mathcal{D}$ are drawn from continuous distribution $\rho(z)$, we can establish the training objective as:

$$\arg \min_{\theta \in \Theta} \mathop{\mathbb{E}}_{z \in \mathcal{D}} [\mathcal{L}(z, \theta)] = \int_z \mathcal{L}(z, \theta) \rho(z) dz. \tag{5}$$

After applying proposed pruning, we sample $z$ according to normalized $(1 - \mathcal{P}_t(z))\rho(z)$. In backpropagation, rescaling loss is equivalent to rescaling the gradient. By rescaling the loss of each sample $z$ with a factor $\gamma_t(z) = 1/(1 - \mathcal{P}_t(z))$ ( $\forall z \in \mathcal{D}, \mathcal{P}_t(z) = 0 \Rightarrow \gamma_t(z) = 1$), the training objective (expanded proof in Appendix B.1) on $\mathcal{S}_t$ becomes:

$$\arg \min_{\theta \in \Theta} \mathop{\mathbb{E}}_{z \in \mathcal{S}_t} [\gamma_t(z) \mathcal{L}(z, \theta)] = \arg \min_{\theta \in \Theta} \frac{1}{c_t} \int_z \mathcal{L}(z, \theta) \rho(z) dz, \tag{6}$$

where $c_t = \mathbb{E}_{z \sim \rho}[1 - \mathcal{P}_t(z)] = \int_z \rho(z)(1 - \mathcal{P}_t(z))dz$, $c_t \in (0, 1)$ is a constant for temporal status $t$. Then the objective in Eqn. 6 is a constant-rescaled version of the original objective in Eqn. 5. Therefore, training on $\mathcal{S}_t$ with rescaled factor $\gamma_t(z)$ could achieve a similar result as training on the original dataset. Furthermore, these operations also leverage the problem of reduced iterations. In real-world applications, considering the dataset as discrete ones, we force the sample number as:

$$\frac{1}{c_t} = \frac{|\mathcal{D}|}{\lfloor \sum_{z \in \mathcal{D}} (1 - \mathcal{P}_t(z)) \rfloor} \simeq \frac{|\mathcal{D}|}{|\mathcal{S}_t|} \Rightarrow \mathbb{E}[\nabla_\theta \mathcal{L}(\mathcal{S}_t)] \simeq \frac{|\mathcal{D}|}{|\mathcal{S}_t|} \mathbb{E}[\nabla_\theta \mathcal{L}(\mathcal{D})]. \tag{7}$$

For each epoch, the iteration number, *i.e.* gradient update number becomes $|\mathcal{S}_t|/|\mathcal{D}|$ of the original one, while our method scale the expected gradient to $|\mathcal{D}|/|\mathcal{S}_t|$. As a result, this would leverage the influence of reduced gradient update number. The approximation will hold when the pruning ratio is not too high and the learning rate is not too big. We provide a detailed analysis in Appendix B.

## 2.4 ANNEALING

Based on the theoretical analysis above, the objectives and updating expectations between InfoBatch and training on the original dataset are approximately the same. However, there still exist minor differences between the optimization on $\mathcal{D}$ and $\mathcal{S}_t$. During training, if a sample is pruned in the middle stage, it is still likely to be revisited afterward. However, in the last few epochs, the revisiting probability drastically drops, resulting in remaining gradient expectation bias. Therefore, given training epoch $C$, we define a ratio hyper-parameter $\delta \in (0, 1)$. The pruning is only conducted in the first $\delta \cdot C$ epochs (in practice, $\delta$ is close to 1 so InfoBatch saves at a reasonable ratio). After that, we train on the full dataset till the end. The corresponding operation can be interpreted as

$$\mathcal{P}_t(z) = \begin{cases} r, & \mathcal{H}_t(z) < \bar{\mathcal{H}}_t \wedge t < \delta \cdot C \\ 0, & \mathcal{H}_t(z) \geq \bar{\mathcal{H}}_t \vee t \geq \delta \cdot C \end{cases}. \tag{8}$$

Combining the above components, InfoBatch achieves lossless training performance with fewer iterations compared with training on the original dataset.

## 3 EXPERIMENTS

### 3.1 DATASETS AND IMPLEMENTATION DETAILS

We verify the effectiveness of our method on multiple datasets: CIFAR-10/100 (Krizhevsky et al., a;b), ImageNet-1K (Deng et al., 2009), ADE20K (Zhou et al., 2017) and FFHQ (Karras et al., 2019).

**Implementation Details.** For InfoBatch, default value $r = 0.5$ and $\delta = 0.875$ are used if not specified. For classification tasks, we train ResNet18 and ResNet-50(He et al., 2016), ViT-Base(MAE) (He et al., 2021) and Swin-Tiny (Liu et al., 2021) for evaluation. On CIFAR-10/100 and ImageNet-1K, all models are trained with OneCycle scheduler (cosine annealing) (Loshchilov & Hutter, 2016; Smith & Topin, 2017) using default setting and SGD/LARS optimizer (You et al., 2017) with momentum 0.9, weight decay 5e-4. All images are augmented with commonly adopted transformations, *i.e.* normalization, random crop, and horizontal flop if not stated otherwise. The implementation is based on PyTorch (Paszke et al., 2019) and Timm (Wightman et al., 2021). On CIFAR100, ImageNet-1K and ADE20K, a more aggressive $r$ (0.75) is utilized for smaller loss samples (20%). For the semantic segmentation task, we conduct experiments on ADE20K (Zhou et al., 2017). The chosen network is UperNet (Xiao et al., 2018) with backbone ResNet-50. We follow the default configuration of the mmsegmentation (Contributors, 2020). All other details can be found in Appendix.

### 3.2 COMPARISONS WITH SOTA METHODS

**Performance Comparisons.** We compare our proposed InfoBatch with static and dynamic data pruning methods in Tab. 1 on CIFAR-10, CIFAR-100. In the first part, we introduce static pruning methods, whose simplest baseline is random selection before training. Influence (Koh & Liang, 2017) and EL2N (Paul et al., 2021) are two classical static pruning methods that prune samples based on Influence-score and EL2N-score. DP (Yang et al., 2023b) conducts pruning with consideration of generalization. To make a wider comparison, we include 10 coreset selection methods. These methods select a coreset of data via their predefined score function or heuristic knowledge. Then, we introduce 3 dynamic pruning methods in the second part. Following (Raju et al., 2021), we also construct a dynamic pruning baseline, termed as Random*, which conducts random selection in each epoch. Compared to Random operation in Tab. 1, the diversity of training samples in Random* is much better than Random. Therefore, as shown in Tab. 1, Random* usually performs better than Random and many static methods. $\epsilon$-greedy (Raju et al., 2021) is inspired by reinforcement learning. UCB (Raju et al., 2021) proposes to prune samples using the upper confidence bound of loss.

Based on the comparisons in Tab. 1, we have the following observations: 1). At 30% pruning ratio, only InfoBatch achieves lossless performances on both datasets. Other methods only obtain near-lossless results on CIFAR-10 dataset while encountering large performance drops on CIFAR-100. 2). As the pruning ratio increases, InfoBatch continuously outperforms other methods by an even larger accuracy margin. It validates the effectiveness of the proposed unbiased dynamic pruning strategy.

**Efficiency Comparisons.** In addition to the performance comparison, we compare the efficiency between Infobatch and other methods. Although the motivations of these methods are diverse, their

Table 1: The accuracy (%) comparison to state-of-the-art methods. All methods are trained with ResNet-18. As InfoBatch has a self-adaptive ratio, we mark the results with $\dagger$ where the same forward propagation number during training is matched by controlling epoch number. Random* denotes dynamic random pruning. Details are available in Appendix A.

| | Dataset | CIFAR10 | | | CIFAR100 | | |
|---|---|---|---|---|---|---|---|
| | Prune Ratio % | 30 | 50 | 70 | 30 | 50 | 70 |
| Static | Random | $94.6_{\downarrow 1.0}$ | $93.3_{\downarrow 2.3}$ | $90.2_{\downarrow 5.4}$ | $73.8_{\downarrow 4.4}$ | $72.1_{\downarrow 6.1}$ | $69.7_{\downarrow 8.5}$ |
| | Herding (Welling, 2009) | $92.2_{\downarrow 3.4}$ | $88.0_{\downarrow 7.6}$ | $80.1_{\downarrow 15.5}$ | $73.1_{\downarrow 5.1}$ | $71.8_{\downarrow 6.4}$ | $69.6_{\downarrow 8.0}$ |
| | Influence (Koh & Liang, 2017) | $93.1_{\downarrow 2.5}$ | $91.3_{\downarrow 4.3}$ | $88.3_{\downarrow 7.3}$ | $74.4_{\downarrow 3.8}$ | $72.0_{\downarrow 6.2}$ | $68.9_{\downarrow 9.5}$ |
| | K-Center (Sener & Savarese, 2018) | $94.7_{\downarrow 0.9}$ | $93.9_{\downarrow 1.7}$ | $90.9_{\downarrow 4.7}$ | $74.1_{\downarrow 4.1}$ | $72.2_{\downarrow 6.0}$ | $70.2_{\downarrow 8.0}$ |
| | DeepFool (Ducoffe & Precioso, 2018) | $95.1_{\downarrow 0.5}$ | $94.1_{\downarrow 1.5}$ | $90.0_{\downarrow 5.6}$ | $74.2_{\downarrow 4.0}$ | $73.2_{\downarrow 5.0}$ | $69.8_{\downarrow 6.4}$ |
| | Forgetting (Toneva et al., 2018) | $94.7_{\downarrow 0.9}$ | $94.1_{\downarrow 1.5}$ | $91.7_{\downarrow 3.9}$ | $75.3_{\downarrow 2.9}$ | $73.1_{\downarrow 5.1}$ | $69.9_{\downarrow 8.3}$ |
| | EL2N-2 (Toneva et al., 2018) | $94.4_{\downarrow 1.2}$ | $93.2_{\downarrow 2.4}$ | $89.8_{\downarrow 5.8}$ | $74.1_{\downarrow 4.1}$ | $71.0_{\downarrow 7.2}$ | $68.5_{\downarrow 9.7}$ |
| | EL2N-20 (Toneva et al., 2018) | $95.3_{\downarrow 0.3}$ | $\mathbf{95.1}_{\downarrow 0.5}$ | $91.9_{\downarrow 3.7}$ | $77.2_{\downarrow 1.0}$ | $72.1_{\downarrow 6.1}$ | - |
| | Least Confidence (Coleman et al., 2019) | $95.0_{\downarrow 0.6}$ | $94.5_{\downarrow 1.1}$ | $90.3_{\downarrow 5.3}$ | $74.2_{\downarrow 4.0}$ | $72.3_{\downarrow 5.9}$ | $69.8_{\downarrow 8.4}$ |
| | Margin (Coleman et al., 2019) | $94.9_{\downarrow 0.7}$ | $94.3_{\downarrow 1.3}$ | $90.9_{\downarrow 4.7}$ | $74.0_{\downarrow 4.2}$ | $72.2_{\downarrow 6.0}$ | $70.2_{\downarrow 8.0}$ |
| | CD (Agarwal et al., 2020) | $95.0_{\downarrow 0.6}$ | $94.3_{\downarrow 1.3}$ | $90.8_{\downarrow 4.8}$ | $74.2_{\downarrow 4.0}$ | $72.3_{\downarrow 5.9}$ | $70.3_{\downarrow 7.9}$ |
| | Craig (Mirzasoleiman et al., 2020) | $94.8_{\downarrow 0.8}$ | $93.3_{\downarrow 3.3}$ | $88.4_{\downarrow 7.2}$ | $74.4_{\downarrow 3.8}$ | $71.9_{\downarrow 6.3}$ | $69.7_{\downarrow 8.5}$ |
| | GraNd-4 (Paul et al., 2021) | $95.3_{\downarrow 0.3}$ | $94.6_{\downarrow 1.0}$ | $91.2_{\downarrow 4.4}$ | $74.6_{\downarrow 3.6}$ | $71.4_{\downarrow 6.8}$ | $68.8_{\downarrow 9.4}$ |
| | Glister (Killamsetty et al., 2021b) | $95.2_{\downarrow 0.4}$ | $94.0_{\downarrow 1.6}$ | $90.9_{\downarrow 4.7}$ | $74.6_{\downarrow 3.6}$ | $73.2_{\downarrow 5.0}$ | $70.4_{\downarrow 7.8}$ |
| | DP (Yang et al., 2023b) | $94.9_{\downarrow 0.7}$ | $93.8_{\downarrow 1.8}$ | $90.8_{\downarrow 4.8}$ | $77.2_{\downarrow 1.0}$ | $73.1_{\downarrow 5.1}$ | - |
| Dynamic | Random* | $94.8_{\downarrow 0.8}$ | $94.5_{\downarrow 1.1}$ | $93.0_{\downarrow 2.6}$ | $77.3_{\downarrow 0.9}$ | $75.3_{\downarrow 2.9}$ | - |
| | Full Data Reduced Epoch | $94.8_{\downarrow 0.8}$ | $94.6_{\downarrow 1.0}$ | $92.7_{\downarrow 2.9}$ | $77.0_{\downarrow 1.2}$ | $76.9_{\downarrow 1.3}$ | $76.3_{\downarrow 1.9}$ |
| | $\epsilon$-greedy (Raju et al., 2021) | $95.2_{\downarrow 0.4}$ | $94.9_{\downarrow 0.7}$ | $94.1_{\downarrow 1.5}$ | $76.4_{\downarrow 1.8}$ | $74.8_{\downarrow 3.4}$ | - |
| | UCB (Raju et al., 2021) | $95.3_{\downarrow 0.3}$ | $94.7_{\downarrow 0.9}$ | $93.9_{\downarrow 1.7}$ | $77.3_{\downarrow 0.9}$ | $75.3_{\downarrow 2.9}$ | - |
| | InfoBatch | $\mathbf{95.6}_{\uparrow 0.0}$ | $\dagger\mathbf{95.1}_{\downarrow 0.5}$ | $\dagger\mathbf{94.7}_{\downarrow 0.9}$ | $\mathbf{78.2}_{\uparrow 0.0}$ | $\dagger\mathbf{78.1}_{\downarrow 0.1}$ | $\dagger\mathbf{76.5}_{\downarrow 1.7}$ |
| | Whole Dataset | | $95.6_{\pm 0.1}$ | | | $78.2_{\pm 0.1}$ | |

Table 2: Comparison of performance and time cost on ImageNet-1K. Results are reported with ResNet-50 under 40% prune ratio for 90 epochs on an 8-A100-GPU server. "Time" is wall clock time; "Total (n*h)" is the total node hour.

| | GC | EL2N-20 | UCB | Ours | Full Data |
|---|---|---|---|---|---|
| Acc (%) | $73.6_{\pm 0.4}$ | - | $75.8_{\pm 0.3}$ | $\mathbf{76.5}_{\pm 0.2}$ | $76.4_{\pm 0.2}$ |
| Time (h) | 10.5 | 10.5 | 10.5 | 10.5 | 17.5 |
| Overhead (h) | >24 | >17.5 | 0.03 | **0.0028** | 0.0 |
| Total (n*h) | >108 | >224 | **84** | **84** | 140.0 |

Table 3: Experiments on ImageNet-1K. ViT-Base(MAE) is pretrained with InfoBatch for 300 epochs and fine-tuned 100 epochs. Swin-Tiny is trained from scratch InfoBatch.

| Model # | Prune Ratio | Orignial | InfoBatch |
|---|---|---|---|
| R-50$_\text{PyTorch}$ | 40.5% | 76.4 | $76.5_{\uparrow 0.1}$ |
| R-50$_\text{Timm}$ | 20.3% | 78.4 | $78.3_{\downarrow 0.1}$ |
| Swin-T | 19.0% | 81.5 | $81.4_{\downarrow 0.1}$ |
| ViT-B(MAE) | 24.8% | 82.8 | $82.9_{\uparrow 0.1}$ |

main goal is to save training costs. Thus, we report training time, extra cost, and total GPU hours of these methods in Tab. 2. Under the same computational condition, static pruning methods GC and EL2N require extra preprocessing time even longer than the actual training run to process the pruning. Previous state-of-the-art dynamic data pruning method UCB (Raju et al., 2021) shows far better efficiency in the pruning process, while still not comparable to InfoBatch. InfoBatch further reduces the extra time cost to $1/10$ of UCB, taking only 10 seconds for 90 epochs of pruning. More importantly, InfoBatch achieves lossless performance on ImageNet-1K at the pruning ratio of 40%.

## 3.3 IMAGENET-1K RESULTS

As InfoBatch is proposed to scale data pruning to large-scale datasets, we train ResNet-50, ViT-Base(MAE) (He et al., 2021), Swin-Tiny (Liu et al., 2021) on ImageNet-1K. The results are shown in Table 3. Evidently, InfoBatch is able to achieve lossless performance on both CNN-based networks and Transformer-based networks with substantial cost savings. The experimental result with Timm (Wightman et al., 2021) indicates that InfoBatch is compatible with existing acceleration methods like mixed-precision training and augmentation/regularization methods including Random Erase, MixUp, CutMix (Zhong et al., 2017; Zhang et al., 2018; Yun et al., 2019) etc. (see appendix A.4).

Table 4: Ablation of proposed operations in the proposed framework on CIFAR-100. We set $r = 0.5$ in the experiments. Random* is same as Tab. 1.

| Operation | | | Acc. | |
|---|---|---|---|---|
| $\mathcal{P}$ | Res | Ann | R-18 | R-50 |
| Random* | | | 77.3±0.2 | 79.7±0.2 |
| Soft | | | 77.5±0.2 | 79.9±0.2 |
| Soft | ✓ | | 78.1±0.2 | 80.3±0.3 |
| Soft | | ✓ | 77.7±0.2 | 80.0±0.2 |
| Soft | ✓ | ✓ | **78.2**±0.2 | **80.6**±0.2 |
| Full Dataset | | | 78.2±0.2 | 80.6±0.1 |

Table 5: Comparison of pruning conditions. Random here is using rescaling and annealing, adjusting the r value to control the pruning ratio. Experiments are conducted on CIFAR-100 R-50.

| Prune Condition | Acc. (%) | Prune. (%) |
|---|---|---|
| $\mathcal{H}_t(z) < \bar{\mathcal{H}}_t$ | 80.6±0.2 | 33 |
| $\mathcal{H}_t(z) > \bar{\mathcal{H}}_t$ | 80.5±0.2 | 16 |
| Random | 80.5±0.1 | 16 |
| Random | 80.5±0.2 | 33 |
| Full Dataset | 80.6±0.1 | - |

The overhead of InfoBatch is visualized in Fig. 3d. Results are suggesting that InfoBatch can greatly improve training costs on various computer vision tasks using large-scale datasets like ImageNet-1K, with a negligible overhead compared to its acceleration.

## 3.4 ABLATION EXPERIMENTS

We perform extensive ablation experiments to illustrate the characteristics of InfoBatch. If not stated, the experiments are conducted on CIFAR-100 by default.

**Evaluating the Components of InfoBatch.** We design an ablation study to investigate the soft pruning policy, expectation rescaling, and annealing operations in Tab. 4. Dynamic random pruning, serving as the baseline method, fails to achieve lossless performance on all the architectures. It can be explained that due to pruning, the gradient update number is reduced compared to training on the original dataset. Similarly, only applying soft pruning obtains marginally better results than dynamic random pruning as expected. In contrast, the proposed rescale and anneal operations are consistently complementary and achieve lossless performances in both settings, which aligns with our theoretical analysis. Moreover, rescaling obtains better correction to the gradient expectation bias on these two tasks than annealing. Using rescaling alone may sometimes achieve lossless results but with higher performance variance and lower mean performance; annealing improves the stability of rescaling. Annealing alone doesn't improve performance significantly.

**Exploring Where to Prune.** In default setting, samples with $\mathcal{H}_t(z) < \bar{\mathcal{H}}_t$ are pruned. Other possible rules are to prune samples with $\mathcal{H}_t(z) > \bar{\mathcal{H}}_t$ or totally random. We compare the performance and cost of these different prune strategies in Tab. 5. Compared to training on the original dataset, the two other strategies also achieve near-lossless performance, while the default pruning condition is better. $\mathcal{H}_t(z) > \bar{\mathcal{H}}_t$ has a lower pruning ratio because of the skewness of loss distribution (visualized in Appendix E). The ablation results align with the theoretical analysis that 1. rescaling and annealing can contribute to keeping the performance; 2. pruning low-loss samples is better because it risks less of potential scaling limit. More discussion is in Appendix B.

**Evaluation of Pruning Ratio.** We define $r$ as the probability to prune a sample $x$ when $\mathcal{H}_t(z) < \bar{\mathcal{H}}_t$. In Fig. 3a, we evaluate different pruning ratios for ResNet-18 (R-18) and ResNet-50 (R-50) on CIFAR-100. Setting $r \leq 0.5$ obtains lossless performance on both architectures, which indicates InfoBatch is relatively robust on hyper-parameter $r$. Another finding is that further increase $r(\geq 0.6)$ leads to degraded performances. This can primarily be attributed to the increased variance and reduced steps when too many samples are pruned. Considering the efficiency and performance, we set $r = 0.5$ by default. Another benefit is that this would not affect float number precision.

**Evaluation of Annealing Ratio.** To reduce the remaining gradient expectation bias, we introduced an annealing operation in the last epochs. $\delta$ represents a quantile of given training epoch number $C$. We evaluate it from 0.75 to 0.95 and report the performances in Fig. 3b. A larger $\delta$ means fewer annealing epochs, leaving more remaining gradient bias, which may result in degraded average performance and increased performance variance. When $\delta \leq 0.875$, lossless performance can be achieved with maximized overall cost saving. Limited tuning effort is required to obtain lossless results.

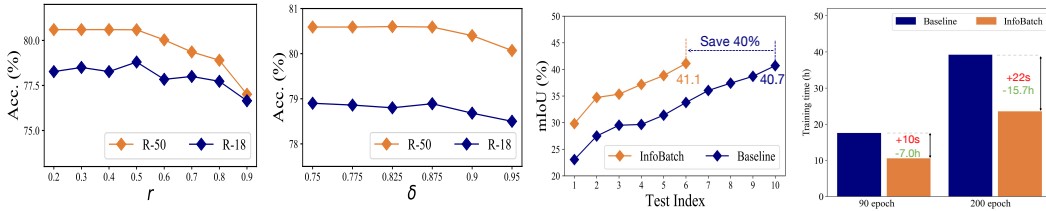

(a) Acc. vs pruning ratio.  (b) Acc. vs annealing ratio.  (c) mIoU on ADE20K.  (d) Overhead and savings.

Figure 3: (a)(b) Evaluation curves of hyper-parameter $r$ and $\delta$ on R-18 and R-50. (c) With 60% iteration and overall cost, InfoBatch achieves lossless performance on ADE20K semantic segmentation. (d) InfoBatch saves the overall cost by 40% on IN-1K. + means costs and - means savings.

Table 6: Cross-architecture robustness results of Info-Batch. 'Full Dataset' denotes training on the original dataset without pruning.

|  | CIFAR-10 | | CIFAR-100 | | ImageNet-1K | |
|---|---|---|---|---|---|---|
|  | R-18 | R-50 | R-18 | R-50 | R-18 | R-50 |
| Full Dataset | 95.6 | 95.6 | 78.2 | 80.6 | 70.5 | 76.4 |
| **InfoBatch** | 95.5 | 95.6 | 78.2 | 80.6 | 70.4 | 76.5 |
| Saved (%) | 41.3 | 41.3 | 33.8 | 41.3 | 26.1 | 40.9 |

Table 7: Comparison of accuracy (%) and saved cost (%) on CIFAR-10 when trained with R-50 using different optimizers. All the results are obtained from the same hardware.

|  | SGD | AdamW | LARS | LAMB |
|---|---|---|---|---|
| Full Dataset | 95.6 | 94.3 | 95.5 | 95.0 |
| InfoBatch | 95.6 | 94.4 | 95.5 | 95.0 |
| Saved (%) | 41.3 | 41.2 | 41.3 | 41.2 |

## 3.5 GENERALIZATION EVALUATION

**Cross-architecture Robustness Evaluation.** InfoBatch is not correlated to specific model architectures, thereby it is a model-agnostic framework. Beyond reported ViT-based results in Tab. 3, we train R-18 and R-50 on CIFAR-10, CIFAR-100, and ImageNet-1K. As illustrated in Tab. 6, we achieve lossless training performances under all settings, which indicates strong generality of InfoBatch. We also report the overall cost saving in Tab. 6. One can observe that InfoBatch can save 26% ~40% overall cost on training these datasets.

**Cross-optimizer Robustness Evaluation.** In the deep-learning area, there are various optimizers (Bottou et al., 1991; Kingma & Ba, 2014; You et al., 2017; 2019; Loshchilov & Hutter, 2019) that are broadly adopted. We verify InfoBatch's robustness across these optimizers. As shown in Tab. 7, InfoBatch achieves lossless performance with four popular optimizers, and the overall cost saving is stable across optimizers. Specifically, we apply InfoBatch to large-batch training optimizers LARS and LAMB. As InfoBatch accelerates in the data dimension, InfoBatch further speeds up training by 1.67 times without extra cost or performance drop. It shows that InfoBatch has the potential to be combined with acceleration methods in other dimensions.

**Cross-task Robustness Evaluation.** To verify the generality of InfoBatch on different tasks other than classification, we apply InfoBatch to the semantic segmentation task. We train UperNet on ADE20K and report the mIoU curves in Fig. 3c. Our implementation is based on mmsegmentation (Contributors, 2020). InfoBatch uses 60% of original iterations to achieve lossless performance (mIoU: InfoBatch 41.12%, original 40.7%). We further apply InfoBatch in the training of Latent Diffusion (Rombach et al., 2021) on FFHQ as demonstrated in Tab. 8 (and generated images in Appendix E). It saves the pretraining cost of Latent Diffusion by ~27%, and the image quality is not affected. The result indicates InfoBatch's generality across segmentation and generation tasks.

Table 8: Diffusion.

| Method | FID |
|---|---|
| Original | 7.83 |
| InfoBatch | 7.70 |

Beyond computer vision tasks, InfoBatch can also be applied to instruction fine-tuning task. We utilize InfoBatch to accelerate LLaMA (Touvron et al., 2023) instruction fine-tuning on Alpaca (Taori et al., 2023) (originally 52k samples). To further reduce training costs, we

Table 9: InfoBatch on instruction tuning with LLaMA-7B.

| Method | Time(m) | BBH | DROP | MMLU | Human-Eval | Avg. |
|---|---|---|---|---|---|---|
| DQ+InfoBatch | 14.4 | 31.0 | 27.5 | 35.3 | 11.6 | 26.3 |
| DQ (2% data) | 18.0 | 32.9 | 27.6 | 36.6 | 8.5 | 26.3 |

utilize dataset quantization (Zhou et al., 2023) to select a subset of instructions (1k samples). The results are demonstrated in Tab. 9. It is shown that InfoBatch is able to accelerate language model instruction finetuning tasks and is compatible with static dataset compression methods.

## 4    RELATED WORKS

In this section, we discuss the related works in static data pruning and dynamic data pruning. These works are most related to InfoBatch. Other related works like dataset distillation, and large batch training are discussed in Appendix C.

**Static Data Pruning.** The motivation for static data pruning methods is to utilize fewer samples while achieving comparable results as the original dataset. Almost all the methods are based on predefined or heuristic metrics. These metrics can be roughly divided into the following categories: geometry-based (Sener & Savarese, 2018; Agarwal et al., 2020), uncertainty-based (Coleman et al., 2019), error-based (Toneva et al., 2018; Paul et al., 2021), decision-boundary-based (Ducoffe & Precioso, 2018), gradient-matching (Mirzasoleiman et al., 2020; Killamsetty et al., 2021a), bilevel optimization (Killamsetty et al., 2021b) and submodularity-based methods (Iyer et al., 2021). Contextual Diversity (CD) (Agarwal et al., 2020), Herding (Welling, 2009), and k-Center remove the redundant samples based on their similarity to the rest of the data. Cal (Margatina et al., 2021) and Deepfool (Ducoffe & Precioso, 2018) select samples based on their difficulties for learning. FL (Iyer et al., 2021) and Graph Cut (GC) (Iyer et al., 2021) consider the diversity and information simultaneously by maximizing the submodular function. GraNd and EL2N (Paul et al., 2021) propose to estimate sample importance with gradient norm and error-L2-norm. AL (Park et al., 2022) propose to use active learning methods to select a coreset. Moderate (Xia et al., 2023) proposed to use the median of different scores as a less heuristic metric. Coverage-centric Coreset Selection (Zheng et al., 2023) additionally considers distribution coverage beyond sample importance. The main limitations of these works can be summarized as 1). The predefined or heuristic metrics hardly work well across architectures or datasets. 2). The extra cost of these methods is not negligible. As illustrated in Tab. 2, the overhead of EL2N is even longer than the training time on ImageNet-1K.

**Dynamic Data Pruning.**    Dynamic data pruning aims to save the training cost by reducing the number of iterations for training. The pruning process is conducted during training and sample information can be obtained from current training. (Raju et al., 2021) proposes two dynamic pruning methods called UCB and $\epsilon$-greedy. An uncertainty value is defined and the estimated moving average is calculated. Once for every pruning period, $\epsilon$-greedy/UCB is used to select a given fraction of the samples with the highest scores and then trains on these samples during the period. Under this dynamic pruning setting, it achieves a favorable performance compared to static pruning methods on CIFAR-10/100(Krizhevsky et al., a;b), while saving the overhead of assessing samples and pruning before training. He et al. (2023) share a similar framework, while taking the dynamic uncertainty as score. Mindermann et al. (2022) propose Reducible Holdout Loss Selection which prioritizes samples neither too easy nor too hard. It emphasizes training learnable samples. Two of the works (Raju et al., 2021; He et al., 2023) adopt sorting operation (cost $O(logN)$ per sample on dataset size N) for obtaining samples with the highest scores, which could be an overhead for extra-large datasets (e.g. ImageNet-1K has 1.28 million pictures) as it is called multiple times.

## 5    CONCLUSION

We present ***InfoBatch***, a novel framework for lossless training acceleration by unbiased dynamic data pruning. InfoBatch shows its strong robustness on various tasks and datasets, achieving lossless training acceleration on classification, segmentation, vision pertaining, and instruction finetuning. InfoBatch reduces the extra overhead by at least 10 times compared to previous state-of-the-art methods, thus being practical for real-world applications. We provide extensive experiments and theoretical analysis in this paper and hope it can help the following research in this area.

**Limitations and Future Work.** 1. Removing samples may cause bias in model predictions. It is advisable to consider this limitation when applying InfoBatch to ethically sensitive datasets. Currently, we haven't found obvious evidence of bias. We will report publicly if we find any. 2. The current version of InfoBatch relies on multi-epoch training schemes. However, GPT-3 (Brown et al., 2020) and ViT-22B (Dehghani et al., 2023) usually train with limited epochs. InfoBatch may need further adaptation on these tasks. We are going to explore new strategies for these tasks in the future.

**Acknowledgements.** This work is supported by Alibaba Group through Alibaba Innovative Research Program. This work is supported by the National Research Foundation, Singapore under its AI Singapore Programme (AISG Award No: AISG2-PhD-2021-08-008). Yang You's research group is being sponsored by NUS startup grant (Presidential Young Professorship), Singapore MOE Tier-1 grant, ByteDance grant, ARCTIC grant, SMI grant (WBS number: A-8001104-00-00), Alibaba grant, and Google grant for TPU usage. We thank Yong Liu, Wangbo Zhao, Zelin Zang, and Zirui Zhu for valuable discussions and feedback.

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

## A    MORE DETAIL OF EXPERIMENTS

We further demonstrate the details of experiments here. The OncCycle Smith & Topin (2017) scheduler (with cosine annealing) has a learning rate curve as Fig. 4

In Tab. 1, the CIFAR-10 experiment can be reproduced with different optimizers: LARS use a max learning rate 2.3 for the OneCycle scheduler under the batch size of 128, and a maximum learning rate of 5.62 for a batch size of 256. SGD using a maximum learning rate of 0.2 under the batch size of 128 achieves a similar result. For the CIFAR-100 experiment with R-18, SGD is used with a max learning rate of 0.03 and batch size of 128 for baseline; due to reduced steps, InfoBatch uses a learning rate of 0.05 in this setting.

Tab. 2 is implemented based on Pytorch/examples. Lars optimizer and a maximum learning rate of 6.4 is used for batch-size 1024 on ImageNet-1K experiments.

In Table. 3/4/5, CIFAR100 R-50 trained with LARS uses batch size 256 and its corresponding maximum learning rate is 5.2.

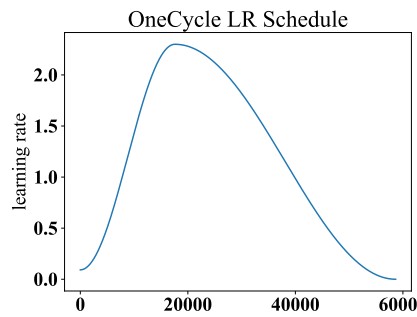

Figure 4: The OneCycle learning rate schedule.

Tab. 7 uses the following hyperparameters: 150 epoch, batch size 128. SGD and LARS use the aforementioned maximum learning rate; Adam, LAMB use the learning rate of 0.0006, and 0.01, respectively.

### A.1    DATASETS

**CIFAR-10/100**. Two CIFAR datasets consist of $32 \times 32$ size colored natural images divided into 10 and 100 categories, respectively. Both use 50,000 images for training and 10,000 images for testing.

**ImageNet-1K** is the subset of the ImageNet-21k dataset with 1,000 categories. It contains 1,281,167 training images and 50,000 validation images.

**ADE20K** is a popular semantic segmentation benchmark. It contains more than 20,000 images with pixel-level annotations of 150 semantic categories.

**FFHQ** is a high-quality image dataset of human faces, contains 70,000 images.

### A.2    DETAIL OF SEMANTIC SEGMENTATION EXPERIMENT

In the semantic segmentation task, there are two types of pixel-level Cross Entropy loss, one for segmentation and one for classification. In backward propagation, the two types of losses are averaged separately at the batch level, then weighted and summed. InfoBatch uses a single score for choosing samples to prune, so we take a different strategy. We use the weighted sum of the two losses at the sample level as the corresponding score. Thereby one can obtain the relative order of sample contribution to training.

The implementation is based on mmsegmentation. As the official code uses step number to configure the training length, we reduce step number from 80k to 48k and set $\delta = 0.85$ to approximate a similar annealing quantile as in other experiments.

### A.3    TINYIMAGENET EXPERIMENT

The Tiny-ImageNet (Le & Yang, 2015) experiment uses batch size 128, image resize 64, SGD as the optimizer, and a learning rate of 0.1 at the start of cosine annealing. InfoBatch also achieves a lossless result on Tiny-ImageNet with ResNet-50. Since most previous works didn't report on Tiny-ImageNet, and Yang et al. (2023b) use a baseline of 51.1 which we cannot reproduce, we report our results here in Tab 10. InfoBatch achieves lossless performance at a saving ratio 41.3%.

## A.4 TIMM EXPERIMENTS DETAIL

Timm is a package including advanced training techniques of deep learning. State-of-the-art results can be obtained by training neural networks with it. To verify the compatibility of our method and advanced training techniques, we apply InfoBatch together with Timm to train ResNet-50 and Swin-Tiny on ImageNet-1K. We further contribute to a method to score samples with strong augmentations like MixUp/CutMix.

Table 10: Tiny-ImageNet Result

|           | Acc. % | Save % |
|-----------|--------|--------|
| Baseline  | 63.5   | -      |
| InfoBatch | 63.4   | 41.3   |

For Swin-Tiny training, we use the following hyperparameters: total epoch 350, linear warmup 10 epoch from 1e-6 to 20e-4, optimizer AdamW, auto augmentation, weight decay 0.05, label smoothing 0.1, image size 224, drop-path 0.2, cutmix alpha 1.0, random erase with pixel model and probability 0.3, cosine annealing to learning rate 1e-6. We also warmup InfoBatch for 5 epochs, only recording scores without pruning.

MixUp/CutMix are two augmentation methods that are not compatible with existing scoring methods: the two augmentation methods would mix the labels and content of images so that the direct loss or other metrics don't reflect a "per-sample" score but a mixture of two sample's score. To reconstruct the per-sample score, we utilize a modified score and estimate the learning progress of a sample as follows:

- 1. In MixUp/CutMix, with a batch-level implementation, images and labels are mixed as $y_{i-mix} = \alpha y_i + (1 - \alpha)y_j, y_{j-mix} = \alpha y_j + (1 - \alpha)y_i, i + j = batchsize - 1$ and $\alpha$ is set for each batch. We take $P_i[y_j]$ as the class-wise score where $P_i$ is the probability vector predicted by the model for mixed sample i.

- 2. To reconstruct a score estimate for each sample, we utilize $score_i = P_{i-mix}[y_i] + P_{j-mix}[y_i]$ and similar for $score_j$.

The idea is that MixUp/CutMix creates linear interpolation between samples, therefore when samples from two classes are mixed, a well-trained classifier should predict the mixed class just as the mixed label, and the probability error from the source class can be traced back. When two samples are from the same class, one can distinguish the source of entropy by linear algebra if $\alpha \neq 0.5$, while it is only about 0.1% chance to mix two samples from the same class for ImageNet-1K.

## A.5 VIT-BASE(MAE) EXPERIMENT DETAIL

We train ViT-Base on ImageNet-1K, based on the implementation of He et al. (2021). We train for 300 epochs with 40 epoch warmup, per GPU batch size 128 with 8 GPU, base linear rate 1.5e-4, mast ratio 0.75, and weight decay 0.05.

## A.6 DIFFUSION MODEL EXPERIMENT DETAIL

Our implementation is based on the public GitHub repo of Rombach et al. (2021). We use V100 for this experiment, due to memory limit, we use half batch size and thus half learning rate according to McCandlish et al. (2018). On FFHQ, InfoBatch uses $\delta = 0.825$ and all other settings as default. Public code didn't provide the implementation of FID calculation so we refer to Seitzer (2020) at link.

## A.7 LLAMA INSTRUCTION-FINETUNING EXPERIMENT DETAIL

Our implementation is based on Killamsetty et al. (2021a); Taori et al. (2023); Zhou et al. (2023). The training config is the same as provided by Zhou et al. (2023).

## B  FURTHER DISCUSSION AND ANALYSIS

### B.1  EXPANDED PROOF OF 2.3

For Eqn. 6, a proof is:

$$
\begin{aligned}
&\arg\min_{\theta\in\Theta} \mathbb{E}_{z\in\mathcal{S}_t}\left[\gamma_t(z)\mathcal{L}(z,\theta)\right] \\
&= \arg\min_{\theta\in\Theta} \frac{\int_z (1-\mathcal{P}_t(z))\gamma_t(z)\mathcal{L}(z,\theta)\rho(z)dz}{\int_z (1-\mathcal{P}_t(z))\rho(z)dz} \\
&= \arg\min_{\theta\in\Theta} \frac{\int_z \mathcal{L}(z,\theta)\rho(z)dz}{\int_z (1-\mathcal{P}_t(z))\rho(z)dz} \\
&= \arg\min_{\theta\in\Theta} \frac{1}{c_t}\int_z \mathcal{L}(z,\theta)\rho(z)dz,
\end{aligned}
\tag{9}
$$

For step 2 expansion, one can refer to the *Importance Sampling* for sampling on a distribution.

### B.2  ON A HARD PRUNE

Hard pruning does not provide enough guarantee on gradient expectation. There is a contradiction about it. From the gradient expectation perspective, only if the hard-pruned samples formulate the same gradient expectation direction as the full dataset will the gradient expectation after pruning be unbiased. It is formulated as follows:

$$
\begin{aligned}
&\mathbb{E}_{z\in\mathcal{S}}[\nabla\mathcal{L}(z)] = c_1 \mathbb{E}_{z\in\mathcal{D}}[\nabla\mathcal{L}(z)] \\
&\Rightarrow \mathbb{E}_{z\notin\mathcal{S}}[\nabla\mathcal{L}(z)] = \frac{\sum_{z\in D\backslash S}\nabla\mathcal{L}(z)}{|\mathcal{D}|-|S|} \\
&= \frac{|\mathcal{D}|\,\mathbb{E}_{z\in\mathcal{D}}[\nabla\mathcal{L}(z)] - c_1|\mathcal{S}|\,\mathbb{E}_{z\in\mathcal{D}}[\nabla\mathcal{L}(z)]}{|\mathcal{D}|-|S|} \\
&= c_2 \mathbb{E}_{z\in\mathcal{D}}[\nabla\mathcal{L}(z)]
\end{aligned}
\tag{10}
$$

$c_1, c_2$ are positive values. Therefore in a hard pruning setting, if assuming the gradient expectation direction is unbiased, the pruned part should also be a valid compression of the original dataset. However, this could contradict some adopted assumptions about pre-defined scores used in previous hard pruning methods. For example, lowest-score samples are often supposed to contribute less to training; but if the resulted gradient expectation is unbiased, pruning them will result in the same result as only using them for training. In contrast, our proposed soft pruning avoids this bias through introducing randomness.

### B.3  FURTHER DISCUSSION ON ANNEALING

Only considering the gradient expectation direction, InfoBatch is unbiased. However, pruning is conducted before each epoch. After pruning, the $\nabla L(S_t, \theta)$ could still be biased within one epoch (not expectation anymore), even though gradients are rescaled. Thereby when the remaining epoch number is low, expectation itself cannot guarantee an unbiased result. That is the reason to introduce the annealing. This introduced 'bias' is not always bad, but annealing gives better stability.

### B.4  FURTHER ANALYSIS ABOUT INFOBATCH'S INFLUENCE ON VARIANCE

We claimed in the main text that InfoBatch can achieve a similar result as training on the original dataset with fewer steps. Explicitly, the assumption is, during the training, after warm-up, the gradient will be relatively stable so that

$$
\sum_{i=1}^{k} \mathbb{E}_{x\in\mathcal{D}}\left[\nabla L(z,\theta_i)\right] \simeq \sum_{j=1}^{\lfloor k*|S_t|/|\mathcal{D}|\rfloor} \mathbb{E}_{x\in\mathcal{S}_t}\left[\nabla L(z,\theta_j)\right].
\tag{11}
$$

A similar case is large-batch training(Goyal et al., 2017), which scales the learning rate to catch up with the training progress. Besides the gradient expectation, another change is in the gradient variance. We use $G = \nabla L$ to denote the expectation of original gradients,

$$
\begin{aligned}
Cov[G_{S_t}] &= \mathbb{E}[(G_{S_t} G_{S_t}^T)^2] - \mathbb{E}[G_{S_t} G_{S_t}^T]^2 \\
&= \frac{1}{|S_t|} \sum_{z \in \mathcal{D}} (1 - \mathcal{P}_t(z)) \frac{G_z G_z^T}{(1 - \mathcal{P}_t(z))^2} - \frac{|\mathcal{D}|^2}{|\mathcal{S}_t|^2} G G^T \\
&= \frac{1}{|S_t|} \sum_{z \in \mathcal{D}} \frac{G_z G_z^T}{(1 - \mathcal{P}_t(z))} - \frac{|\mathcal{D}|^2}{|\mathcal{S}_t|^2} G G^T.
\end{aligned}
\tag{12}
$$

The diagonal terms of the matrix are the actual variance of $G_{S_t}$, which is given by

$$
\frac{1}{|S_t|} \sum_{z \in \mathcal{D}} \frac{G_z^2}{(1 - \mathcal{P}_t(z))} - \frac{|\mathcal{D}|^2}{|\mathcal{S}_t|^2} G^2.
\tag{13}
$$

If $\mathcal{P}_t(z)$ is uniform across all samples (random prune), then Eq. 13 becomes

$$
Var[G_{\mathcal{S}_t}] = \frac{|\mathcal{D}|^2}{|\mathcal{S}_t|^2} (\underset{\mathcal{D}}{\mathbb{E}}[G_z^2] - G^2) = \frac{|\mathcal{D}|^2}{|\mathcal{S}_t|^2} Var[G_\mathcal{D}].
\tag{14}
$$

The standard-deviation-to-expectation ratio, which is the noise-to-signal ratio, is unchanged, since both standard deviation and mean is rescaled by $|\mathcal{D}|/|\mathcal{S}_t|$.

When using a policy $\mathcal{H}_t(z) < \bar{\mathcal{H}}_t$, assuming generally lower loss samples has a smaller gradient, then the rescaled gradients are smaller, and we make the following derivation:

$$
\begin{aligned}
\sum_{z \in \mathcal{D}} \frac{G_z^2}{(1 - \mathcal{P}_t(z))} &= |\mathcal{D}| \underset{\mathcal{D}}{\mathbb{E}}[G_z^2] - \sum_{z \in \mathcal{D}} G_z^2 + \sum_{z \in \mathcal{D}} \frac{G_z^2}{(1 - \mathcal{P}_t(z))} \\
&= |\mathcal{D}| \underset{\mathcal{D}}{\mathbb{E}}[G_z^2] + \sum_{z \in \mathcal{D}} \frac{\mathcal{P}_t(z) G_z^2}{(1 - \mathcal{P}_t(z))}.
\end{aligned}
\tag{15}
$$

As we already set $\mathcal{P}_t(z) = 0$ for larger $G_z$ (high loss samples), therefore when assuming $\mathbb{E}_{z|\mathcal{P}_t(z) \neq 0}[G_z^2/(1 - \mathcal{P}_t(z))] \leq \mathbb{E}_\mathcal{D}[G_z^2]$ for rescaling,

$$
\sum_{z \in \mathcal{D}} \frac{\mathcal{P}_t(z) G_z^2}{(1 - \mathcal{P}_t(z))} \leq \sum_{z \in \mathcal{D}} \mathcal{P}_t(z) \underset{D}{\mathbb{E}}[G_z^2] = (|\mathcal{D}| - |\mathcal{S}_t|) \underset{D}{\mathbb{E}}[G_z^2] \leq \frac{|\mathcal{D}|}{|\mathcal{S}_t|} (|\mathcal{D}| - |\mathcal{S}_t|) \underset{D}{\mathbb{E}}[G_z^2],
\tag{16}
$$

where the equity only takes effect at $|\mathcal{D}| = |\mathcal{S}_t|$. Substituting Eq. 16 back to Eq. 15, we get

$$
\sum_{z \in \mathcal{D}} \frac{G_z^2}{(1 - \mathcal{P}_t(z))} \leq |\mathcal{D}| \underset{\mathcal{D}}{\mathbb{E}}[G_z^2] + \frac{|\mathcal{D}|}{|\mathcal{S}_t|} (|\mathcal{D}| - |\mathcal{S}_t|) \underset{D}{\mathbb{E}}[G_z^2] = \frac{|\mathcal{D}|^2}{|\mathcal{S}_t|} \underset{D}{\mathbb{E}}[G_z^2],
\tag{17}
$$

and then substitute it back to Eq. 13,

$$
Var[G_{S_t}] \leq \frac{|\mathcal{D}|^2}{|\mathcal{S}_t|^2} \underset{D}{\mathbb{E}}[G_z^2] - \frac{|\mathcal{D}|^2}{|\mathcal{S}_t|^2} G^2 = \frac{|\mathcal{D}|^2}{|\mathcal{S}_t|^2} Var[G_\mathcal{D}],
\tag{18}
$$

thereby the standard-deviation-to-expectation ratio is likely to be decreased in our pruning condition. The analysis above discussed InfoBatch's influence on variance and signal-to-noise ratio; it also provides insight into the possible future design of dynamic pruning methods with rescaling, that for rescaled samples, $\mathbb{E}_{rescaled}[G_z^2/(1 - \mathcal{P}_t(z))] \leq \mathbb{E}_\mathcal{D}[G_z^2]$ could be a important condition for rescaling.

### B.5 RESCALING LIMIT

Besides the variance discussed above, another key factor to be discussed is the step size. Linear scaling (Goyal et al., 2017) has been proposed in previous research on large batch training, which is observed to have an upper limit when scaling batch size up. The curvature of the loss surface and the noise scale are the two factors affecting the quality of the update. As InfoBatch is using a rescaled gradient, its influence on updates should be considered.

Suppose all samples $z$ are drawn from distribution $\rho$. Denoting the per-sample covariance matrix of original gradient $G_z(\theta) = \nabla_\theta L_z(\theta)$ as $\Sigma(\theta) \equiv cov_{z\sim\rho} G_z(\theta)$, and $G = E_{z\sim\rho} G_z(\theta)$. Taking a second order Taylor expansion at update using SGD optimizer, the loss changes as

$$E[L(\theta - \epsilon G_{S_t})] = L(\theta) - \epsilon E[G_{S_t}^T G] + \frac{1}{2}\epsilon^2 \left( G_{S_t}^T HG_{S_t} + \frac{tr(H\Sigma_{S_t})}{B} \right) \tag{19}$$

$$\simeq L(\theta) - \epsilon \frac{|D|}{|S_t|}|G|^2 + \frac{1}{2}\epsilon^2 \frac{|D|^2}{|S_t|^2} \left( G^T HG + \frac{tr(H\Sigma)}{B} \right) \tag{20}$$

In SGD update, this is taking an effect as using a learning rate of $\epsilon \frac{|D|}{|S_t|}$ instead of $\epsilon$. For the loss to go down, using the original gradient we should have

$$\epsilon < \frac{2|G|^2}{G^T HG + \frac{tr(H\Sigma)}{B}} \tag{21}$$

and now we need

$$\epsilon < \frac{2|G|^2|S_t|}{(G^T HG + \frac{tr(H\Sigma)}{B})|D|}. \tag{22}$$

Note that in the above section B.4 we have analyzed that the pruned variance is likely to be smaller than exact $\frac{|D|^2}{|S_t|^2}$ times of original, which means the diagonal of the covariance matrix, namely the variance vector, is smaller than in equation 20. This could relax this bound when the Hessian is suitable (main weight on diagonal). Still, the curvature-dependent $G^T HG$ term is affected, and one should take care when using a large learning rate where the learning rate is already very close to this bound.

## C  OTHER RELATED WORKS.

**Large batch training** fully exploits the accelerator's parallelism power. Nonetheless, it cannot reduce the overall cost. With fewer iterations, a large learning rate is needed (Goyal et al., 2017; Hoffer et al., 2017) to keep the update size but make the training unstable. LARS (You et al., 2017) and LAMB (You et al., 2019) stabilize the training process by normalizing layer-wise gradients. Stochastic optimization with importance sampling (Zhao & Zhang, 2014; Csiba & Richtárik, 2016; Johnson & Guestrin, 2018) tries to accelerate the optimization convergence speed by sampling certain samples more frequently. A drawback is that the speed-up is sensitive to the model and dataset(Zhao & Zhang, 2014). It cannot directly fit into a training scheme without measurement.

**Dataset distillation.** Our work is also related to dataset distillation (DD), a method that aims to condense the large original datasets into small but informative ones. DD can be mainly divided into the following types: gradient matching methods (Zhao et al., 2021; Liu et al., 2023a; Cui et al., 2023; Yang et al., 2023a; Liu et al., 2023b), distribution matching methods (Zhao & Bilen, 2021; Wang et al., 2022; Sajedi et al., 2023), trajectory matching methods (Cazenavette et al., 2022; Du et al., 2023; Guo et al., 2024; Zhang et al., 2024a;b), and generative prior methods (Zhao & Bilen, 2022; Wang et al., 2023; Cazenavette et al., 2023; Zhang et al., 2023; Gu et al., 2024). However, the main drawbacks of the DD method are its poor scalability and efficiency. DD is hard to apply to large-scale datasets, such as ImageNet-1K and ImageNet-22K.

## D  SPEED OF DIFFERENT OPERATIONS

Table 11: Time cost on same hardware of different operations on different dataset sizes in ms.

|  | sort | percentile | mean |
|---|---|---|---|
| ImageNet-1k (1.28M) | 123.4 | 13.0 | 1.2 |
| ImageNet-22k (14M) | 1630.1 | 232.6 | 6.4 |

Theoretically, for an array of length N, getting the mean or certain percentile of it cost O(N) time complexity, and sorting cost O(NlogN). We can see from 11 that at the ImageNet dataset scale, the

time cost of different operations shows a magnitude difference. Using expensive operations for even larger datasets like JFT-300M could incur much more overhead than cheaper operations due to the theoretical log(N) complexity difference and possible degree of parallelism.

# E VISUALIZATIONS

## E.1 INFOBATCH ON ENTROPY-BASED OBJECTIVE

In Fig. 5, we visualize the equivalent loss distribution after pruning and rescaling. The claim of "equivalent" is formulated as follows: In entropy-based optimization, the original objective is the maximum likelihood

$$\arg\max_{\theta} P(\theta|\mathcal{D}). \tag{23}$$

And after applying Bayesian Theorem, it becomes

$$\arg\max_{\theta} P(\mathcal{D}|\theta) = \arg\max_{\theta} \prod_{z\in\mathcal{D}} P(z|\theta). \tag{24}$$

Take the negative log, the above equation becomes

$$\arg\min_{\theta} \sum_{z\in\mathcal{D}} -logP(z|\theta), \tag{25}$$

which is the entropy-based optimization objective for training. Therefore, multiplying one sample's loss value by a factor is equivalent to duplicating it by the same ratio, from the maximum likelihood perspective.

The loss distribution of CIFAR-10 trained with ResNet50 is shown in the first sub-figure of Fig. 5. Soft pruning randomly prunes samples with loss smaller than the mean as in the second sub-figure in Fig. 5. This leads to insufficient and biased updates, resulting in performance degradation. The rescaling operation of InfoBatch shifts the distribution rightwards. In classification tasks, the Cross-Entropy loss is actually the sum of negative log-likelihood. Therefore, rescaling a sample's loss by $1/(1-r)$ times is

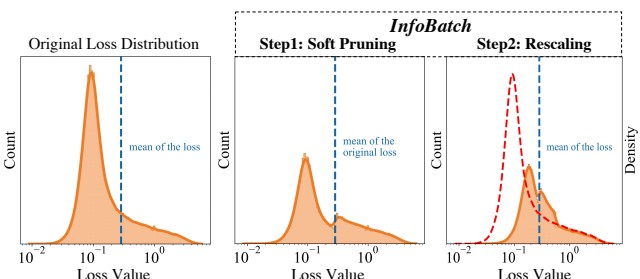

Figure 5: Loss distribution visualizations before and after applying InfoBatch. Best viewed in color.

basically equivalent to duplicating it into $1/(1-r)$ copies. By rescaling, InfoBatch approximates a loss distribution similar to the original one in the first sub-figure. We show the equivalent distribution (the red dashed line) and the actual scaled distribution in the third sub-figure of Fig.5.

In Tab 12, we show the result of training ResNet-50 with full dataset using the same computation as InfoBatch. By reducing the epoch number accordingly, we observe a performance degradation of training on the full dataset. This indicates InfoBatch has better data efficiency.

Table 12: InfoBatch has a higher sample efficiency.

|  | CIFAR-10 | CIFAR-100 |
|---|---|---|
| InfoBatch | 95.6 | 80.6 |
| Full Dataset | 95.2 | 79.4 |

In Fig. 6, we plot the pruning fraction for each epoch during the training of ResNet-50 on CIFAR-10 with InfoBatch. It increases with training adaptively. The corresponding loss distribution shift is visualized in Fig 7. The loss is initially left-screwed and finally right-skewed with a long tail. This is a common tendency of Cross-Entropy loss.

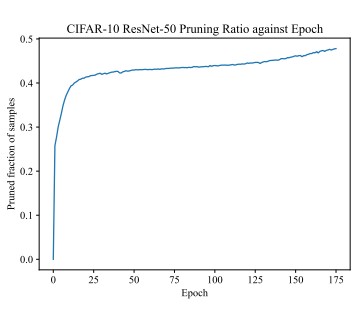

Figure 6: Pruning fraction of mean threshold during training ResNet-50 on CIFAR-10 with InfoBatch. It increases with training adaptively.

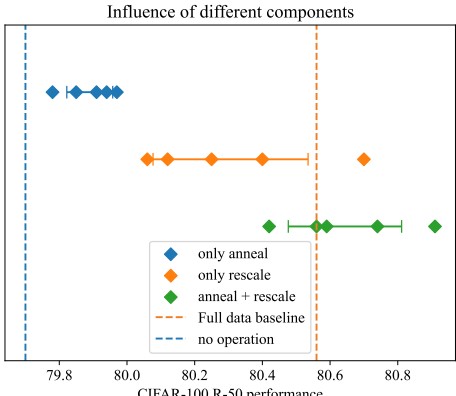

(a) Epoch 10     (b) Epoch 20     (c) Epoch 30

(d) Epoch 50     (e) Epoch 100     (f) Epoch 150

Figure 7: CIFAR-10 ResNet-50 loss distribution over epochs. It has a long-tail tendency.

## E.2 EFFECT OF DIFFERENT COMPONENTS

In Fig. 8, we visualize the training results of ResNet-50 on CIFAR-100 under multiple runs with different components. We can see that with only annealing, the performance is slightly improved over the pruning baseline; with only rescaling (controlling the same pruning ratio with percentile), the performance is increased with a larger variance; with both rescaling and annealing applied, InfoBatch can surpass the full data baseline with stable performance.

## E.3 VISUALIZATION OF DIFFUSION RESULT

In Fig. 9, we visualize some diffusion-model-generated images trained with InfoBatch on the FFHQ dataset. The image quality is as good as trained with uniform random sampling, while the training cost is saved by 27%.

Figure 8: The effect of different components

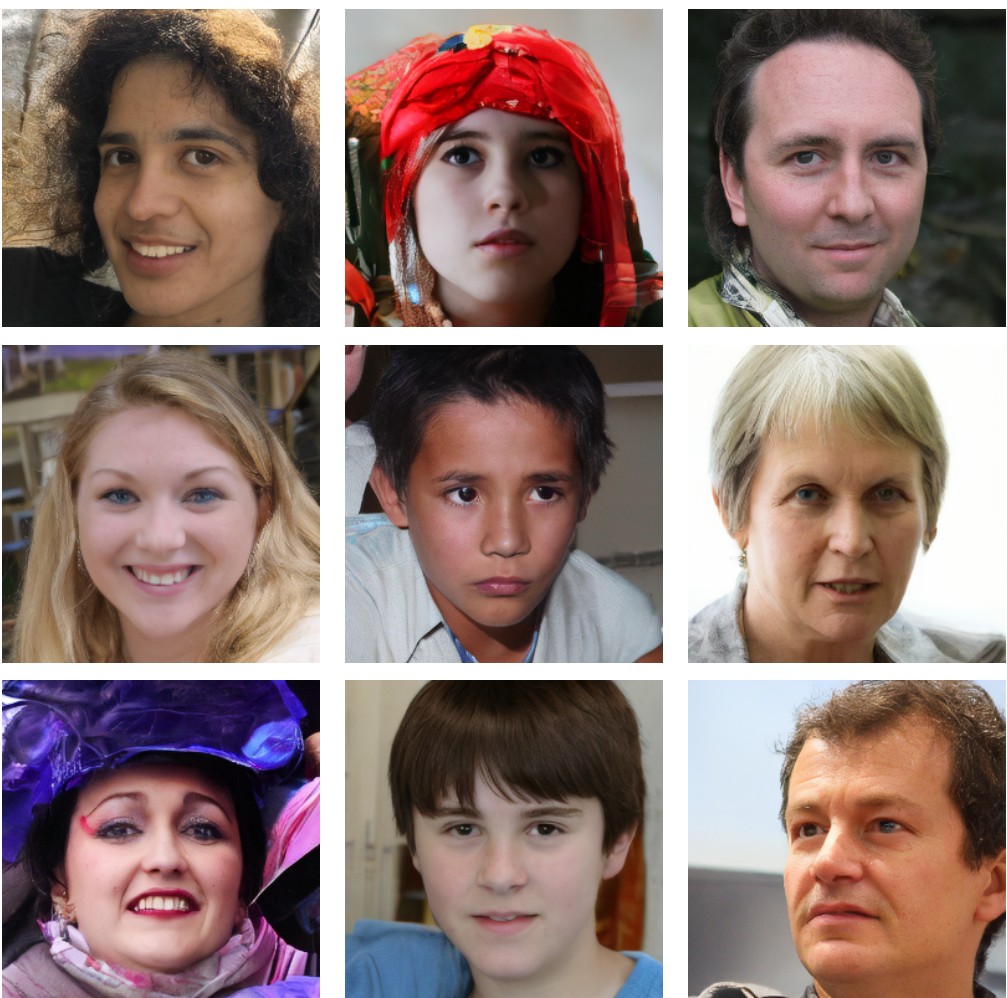

Figure 9: Latent-Diffusion model generated samples trained with InfoBatch on FFHQ. All images are synthesized. We follow the privacy terms of FFHQ dataset.

