# OpenReview forum: "InfoBatch: Lossless Training Speed Up by Unbiased Dynamic Data Pruning"
_ICLR.cc/2024/Conference — ICLR 2024 oral_

### Official Review · Reviewer_DemB · 2023-10-25

**Soundness:** 3 good
**Presentation:** 3 good
**Contribution:** 3 good
**Rating:** 6
**Confidence:** 4

**Summary:**

The authors propose a novel dynamic data pruning, aiming to remove (with a predefined probability) the samples with lower loss score. The algorithm does not require to sort the losses, but it does require to train the model with all samples in the final epochs. The experimental results show a significant speedup in the training procedure, with minimal (or zero) performance drop.

**Strengths:**

**originality**: Although the idea of dynamic pruning is not novel (as clearly stated by the authors), but the solution provided is original enough.

**quality**: The experimental results show the algorithm is able to obtain a clear speedup training, and also remarkable performance results, with almost no accuracy drop.

**clarity**: The idea is simple and easy to implement. The ablation study also provides a solid explanation about how each novel idea affects the overall result.

**significance**: Speeding up training procedures is of huge interest, as it can save a lot of time and energy.

**Weaknesses:**

**originality**: The idea is somehow similar to other dynamic pruning approaches. It does not provide a different point of view in the matter.

**quality**: In the ablation study, I would like to see if different threshold selections (apart from the mean value) can affect the algorithm. I find it a little bit odd that there is no discussion regarding to this point.

**clarity**: The text is too dense. The figures are too small to be read in a paper. I suggest the authors to increase the figures, while removing the text surrounding them. Some sections, like 2.3, can be summarized to make room for the adjustment.

**significance**: the authors claim their threshold value can be established in constant time, whereas the state-of-the-art methods require a sorting part (the complexity should be $\mathcal{O}(N \log N)$. However, I think this improvement is no significant, as the speedup produced is residual compared to the time needed to train the network.

**Questions:**

- Why there is no discussion regarding to the threshold selection procedure? Different solutions like taking the mean plus a factor of the standard deviation can led to interesting results.

- Did the authors experiment with a pruning probability that depends on the sample score?

**Details Of Ethics Concerns:**

I think the authors should address the issue that can cause a bias in the final training, as removing certain samples that can cause this issue.

---

> ### Author Response · Authors · 2023-11-20
> **Response to DemB (1/3)**
>
> We sincerely thank the reviewer DemB for the careful review and valuable comments/questions.
> For the concerns and questions, we make responses as follows.
>
> **Q1: Idea similar to other dynamic pruning approaches, not providing a different point of view in the matter.**
>
> **A1:** Thanks for the comment. In section 1 paragraph 5, we stated:
> * "Directly pruning data may lead to a biased gradient
> estimation as illustrated in Fig. 1a, which affects the convergence result. This is a crucial factor that
> limits their performance, especially under a high pruning ratio (corresponding results in Tab. 1)"
>
> To better illustrate the difference, we show a table here for comprehensive analysis:
>
> | method                       | soft prune | rescale | anneal | unbiased| Task                                         | lossless saving ratio |
> |------------------------------|------------|---------|--------|---------|----------------------------------------------|------------------------|
> | Other Dynamic Pruning [1][2] | no         | no      | no     | no      | Classification                               | ~20%                   |
> | InfoBatch                    | yes        | yes     | yes    | yes     | Classification, Segmentation, Diffusion, LLM | 40%                    |
>
>
> **Analysis:**
> * Previous methods don't use soft pruning. It could lead to a biased gradient direction (see Appendix B.1, or B.2 in the updated revision).
> * Previous methods don't use rescaling. This leads to a lack in total update, which is more severe at a higher pruning ratio.
> * Previous methods don't use annealing. Therefore a remaining bias could be left (Appendix B.2 or updated revision B.3 further discussed annealing).
> * Previous methods mainly focus on classification. InfoBatch can be applied to a broader range of tasks.
>
>
> **Conclusion:**
> InfoBatch proposes a **probabilistic framework** aiming to achieve unbiased dynamic pruning (with soft pruning, rescaling, and annealing), which differs from
> other dynamic pruning works that propose **metrics** and use **sort** to select more important samples.
>
>
> **Q2: Ablation of threshold selection.**
>
> **A2:** Thanks for the comment. We conduct these ablations on CIFAR-100 ResNet-50 as follows:
>
> **Setting:**  All results are averaged across three runs with std reported.
> We report their acc and pruning ratio here.
>
> | threshold      | Acc. (%) | Prune. (%) |
> |----------------|----------|------------|
> | <mean(default) | 80.6±0.1 | 32.8       |
> | <mean+0.5*std  | 80.2±0.1 | 35.4       |
> | <mean+ 1*std   | 80.1±0.1 | 38.2       |
> | <mean+ 2*std   | 80.1±0.1 | 41.5       |
> | <75%           | 80.6±0.1 | 32.8       |
> | <85%           | 80.3±0.1 | 36.9       |
>
> **Analysis:**
> * mean + c*std as a threshold with c>0 can achieve a higher saving ratio but with degraded performance.
> * percentile threshold selection can also lead to a reasonable performance.
> * Specifically, the <75% threshold gets a similar saving ratio and performance as using mean.
> * <85% threshold achieves both higher saving ratio and performance than
> mean+0.5*std
>
> **Discussions:**
> * According to the inference in 2.3, the unbiasedness doesn't depend on the threshold selection. **The experiment result matches the theoretical analysis**
> as using the 75 percentile threshold also gives a lossless result.
> * std factor is not preferred, properly because entropy loss is not Gaussian and long-tailed at convergence
> (we add a visualization in revised Appendix E.1).
> * if assuming Gaussian distribution for some other metrics behind loss, percentile of loss can do the same thing as using means + c*std.
>
> **Conclusion:** InfoBatch is robust to different threshold selection methods. Empirically, the std factor is not preferred for loss. Percentile thresholds could also be a choice.

---

> ### Author Response · Authors · 2023-11-20
> **Response to reviewer DemB (2/3)**
>
> **Q3: Did the authors experiment with a pruning probability that depends on the sample score?**
> **A3:** Thanks for the question. To improve data efficiency, we use the percentile method to prune part of the data (20%) with a more aggressive ratio r (0.75)
> as the default setting on several datasets, as claimed in section 3.1.
>
> By doing so, a higher pruning ratio can be achieved without performance loss. In Table 6 with this improvement,
> CIFAR-100 ResNet-50 can achieve 41.3% saving ratio, compared to 33% in Table 5 with uniform r.
>
> A theoretical related analysis is in Appendix B.3 (updated revision B.4), suggesting that $\mathbb{E}\_\text{rescaled}[G\_{z}^2/(1-\mathcal{P}\_t(z))]\leq {\mathbb{E}}\_{\mathcal{D}}[G_z^2]$
> could be a condition for healthy rescaling. This indicates that we can prune and rescale more aggressively for
> samples with low loss (gradients).
>
> To better illustrate the effect of using different pruning probabilities depending on the sample score,
> we verified different combinations as follows:
>
> **Setting:** InfoBatch on CIFAR-100 ResNet-50 with different thresholding method.
> All results are averaged across three runs with std reported.
> We report their acc and pruning ratio here.
>
> | range:ratio(r)                                                         | saving ratio (%) | performance (%)  |
> |------------------------------------------------------------------------|--------|--------------|
> | full data (r=0)                                                        | 0      | 80.6±0.1     |
> | <mean:0.5                                                              | 33     | 80.6±0.1     |
> | [0%,20%):0.75,[20%,85%):0.5                                            | 41.3   | **80.6±0.2** |
> | [0%,30%):0.75,[30%,90%):0.5                                            | 45     | 80.4±0.2        |
> | [0%,36%):0.75,[36%,95%):0.5                                            | 49     | 80.3±0.2        |
> | [0%,20%):0.75，[20%,40%):0.7，[40%,60%):0.65，[60%,80%):0.6，[80%,90%):0.5 | 51.3   | 80.1±0.2         |
>
>
> **Analysis:**
> * Adaptive r with [0%,20%):0.75,[20%,85%):0.5 gets the 41.3% saving ratio with lossless performance.
> * Adaptive r depending on the score could require a corresponding parameter search.
> * Moving thresholds rightward would cause degraded performance.
>
> **Conclusions:**
> * A pruning probability depends on sample score could further improve data efficiency.
> * It is not straightforward to find a better adaptive r with manually tuning
>
> **Future work:** **To easily find adaptive r values,** we plan to propose a cheap estimator for gradient G and try setting adaptive r based on it ensuring $\mathbb{E}\_\text{rescaled}[G\_{z}^2/(1-\mathcal{P}\_t(z))]\leq {\mathbb{E}}\_{\mathcal{D}}[G_z^2]$.
>
>
> **Q4: Summarize 2.3 and increase the figures.**
>
> **A4:** Thanks for the comments. We shortened the 2.3 and rearranged the layout to enlarge the figures (in the updated revision). The corresponding change
> is as follows:
> * Sec 1, we enlarge Fig 1 and move it to the top of the page.
> * Sec 2.3, we shorten the Theoretical analysis and move part of the proof to B.1.
> * Sec 3, we put the previous Fig 3 and Fig 4 together to the top, making them no longer surrounded by text.
> * Sec 3.5, we increase the size of Table 8 and Table 9.
>
>
> **Q5: Overhead acceleration not significant compared to current dynamic pruning with sort**
>
> **A5:** Thanks for the question. Sort is O(NlogN) for full data and amortized O(logN) for each sample, ours is O(N) and
> amortized O(1). We design the operation considering it could differ in a large factor when N is very big, as shown in Appendix D table 11.
>
> The overhead saving of operation compared to UCB is 10 times (0.03h to 0.0028h) on ImageNet-1K; however we agree it is
> not a significant improvement compared to training time saving (17.5h->10h). Our main saving comes from saving the forward and backward
> computation, which is shown in the table below:
>
> **Setting:** ResNet-50 on CIFAR-100. All results are averaged across three runs with std reported.
> We report their acc and pruning ratio here.
>
> |           | Overhead (s) | Saving      | Acc. (%) |
> |-----------|--------------|-------------|----------|
> | UCB       | < 2          | 20%, ~800s  | 80.4±0.2 |
> | UCB       | < 2          | 33%, ~1320s | 79.8±0.2 |
> | UCB       | < 2          | 41%, ~1640s | 79.5±0.2 |
> | InfoBatch | < 0.1        | 33%, ~1320s | 80.6±0.1 |
> | InfoBatch | < 0.1        | 41%, ~1640s | 80.6±0.2 |
>
> **Analysis**:
> * At lossless performance of ResNet-50 on CIFAR-100, InfoBatch can save 41% cost compared to 20% of UCB. The saved time is a 100% (800s->1600s) improvement.
> * At the same saving ratio, UCB has a much higher performance degradation
>
> We degraded the claim of the part on time complexity, which can be visible in the new revision. The main modifications can
> be summarized as:
> 1. in section 4 related works, we change "which could be a non-negligible overhead" -> "which could be an overhead"

---

> > ### Author Response · Authors · 2023-11-20
> > **Response to reviewer DemB (3/3)**
> >
> > **Q6: Ethics Concerns of removing samples**
> >
> > **A6:** Thanks for the question. InfoBatch uses the full dataset in the final epochs to alleviate the potential risk of
> > gradient bias caused by removing samples.
> > We are not sure whether we fully get the point of the reviewer's concern. We are welcome to more advice and discussion.
> >
> > Here we have some experimental results supporting the unbiasedness. On CIFAR-10 ResNet-50, the orignal per class
> > correct prediction and InfoBatch's per class correct prediction are as follows:
> >
> >
> > | Setting\Classes        |plane| car | bird | cat | deer|  dog | frog | horse | ship | truck |
> > |------------------------|-----|-----|------|-----|-----|------|------|-------|------|-------|
> > | original               |961| 980 | 945  | 908 |  959| 925  | 972  | 970   | 971  | 966   |
> > | InfoBatch (saving 40%) |971| 985 | 940  | 913 | 966| 921  | 983  | 964   | 961  | 969   |
> >
> > **Analysis:** Their cosine similarity is 0.99997, and the predictions match well with no observable bias.
> >
> > **Conclusion:** The result suggests that InfoBatch's design is not prone to the bias caused by removing samples.

---

> > > ### Comment · Reviewer_DemB · 2023-11-20
> > >
> > > Thank you for the response. I suggest the authors to put a disclaimer in the paper related to how data pruning can cause an increase in the bias. More experiments need to be done before concluding that data pruning is not prone to cause bias. Besides that, I am satisfied with the answers.

---

> ### Author Response · Authors · 2023-11-20
> **Response to reviewer DemB on bias (Ethics)**
>
> Thank you for the advice. We agree that ethical issue should be studied with care and not overstated.
> We would evaluate the ethical issue carefully with extended experiments in final revision. We add a disclaimer
> in limitations (in the updated revision), claiming the potential bias should be considered when applying this research:
> * In Section 5 limitations, we add "Removing samples may cause bias in model predictions. It is advisable to consider
> this limitation when applying InfoBatch to ethically sensitive datasets."
>
> We are welcome to further discussions if there are further questions.

---

### Official Review · Reviewer_8u8Q · 2023-10-25

**Soundness:** 3 good
**Presentation:** 3 good
**Contribution:** 3 good
**Rating:** 8
**Confidence:** 4

**Summary:**

The paper proposes a dynamic data pruning scheme to reduce the cost of stochastic gradient training. It does so by stochastically removing lower loss data points and correspondingly rescaling their gradients to maintain an unbiased overall estimates. In conjunction with training on all data towards the end of training, the scheme is able to reduce training cost by about 20-40% on a range of datasets and architectures, including some large scale ones such as ImageNet and a LLaMA model.

Overall, this is a simple but highly pragmatic and practical approach. The scheme solely relies on quantities that are computed during training anyway, so incurs minimal overhead. Unfortunately, I believe that the comparison with the baselines is not entirely applies-to-apples and there are some minor issues with the write-up, so that all things considered I would lean towards rejecting the paper. Nevertheless, I hope these will be addressed over the course of the rebuttal and am open to increasing my score.

EDIT: the extensive additional results for the rebuttal have addressed my concerns and I would now recommend acceptance.

**Strengths:**

* The approach is simple but pragmatic, I appreciate the care that is taken to not incur substantial overheads for additional computation such as thresholding by the mean score. This could be a broadly applicable technique for speeding up training, both for researchers and practitioners.
* The method is described well, I think it would be straight-forward to implement this even without code being provided.
* There is a broad range of experiments, including some larger scale setting involving ImageNet and language models, emphasizing the potential relevance of the approach.

**Weaknesses:**

* The comparison with the baselines does not seem entirely apples-to-apples to me due to the "annealing" period on the whole dataset. I suspect (intuitively and based on the ablations in Table 4 plus the pruning rule seemingly being irrelevant in Table 5) that ensuring the total length of the optimization trajectory remains comparable to that on the full dataset (by rescaling the gradients) in conjunction with the fine-tuning on all data towards the end of training is the "secret sauce" to making a dynamic pruning method perform without a degradation in test accuracy. I'm not familiar with the (Raju et al., 2021) paper, but would expect that at least the annealing period could be incorporated into this method without further issue. At the moment the paper presents its selection rule leading to performance matching that of full-data training as a core contribution, however if this can be achieved relatively easily with other selection techniques as well, I think it becomes more about the re-scaling/tuning as general purpose techniques and the cost comparison between different selection approaches being featured more prominently. I don't think this would worsen the paper at all, although it would change the key takeaways a fair bit and I think it is important that the latter accurately reflect the empirical results.
* On a related note, I am a little bit concerned that the hyperparameters on e.g. ResNets for CIFAR10 are not tuned for achieving the final test performance as quickly as possible. I think it would be worth adding a baseline that trains on all data with a reduced number of epochs/learning rate decay milestones but increased learning rate corresponding to the computation saved by pruning (so hypothetically for 20% saved computation, train for 80 instead of 100 epochs but with learning rate 1.25 instead of 1 and halve it after 40 instead of 80 epochs). This is to ensure that pruning approaches meaningfully speed up training rather than benefitting from slack in the canonical hyperparameter choices for benchmark problems.
* I don't entirely follow what the theoretical analysis is trying to achieve in 2.3. Isn't this just showing that the soft-pruned and rescaled gradient is unbiased? Isn't this completely obvious from having independent Bernoullis multiplied onto the terms of a sum (the total gradient over the dataset) and the expectation of a Bernoulli being its probability (so that if we divide by the probability, we get an expectation of 1 and the sum remains unchanged)?
* I found the paper to be fairly different to what the title made me expect. "Info" and "lossless" imply a connection with information theory and lossless compression to me, which is of course not present in the method. I appreciate that this is entirely subjective, but would suggest reconsidering the title of the paper. In particular, I would argue that the "lossless" part is a bit misleading since this is not theoretically guaranteed by the method, but merely and empirical observation in the experiments. Of course matching performance to the full dataset can always be achieved by letting $r \rightarrow 0, \delta \rightarrow 1$, but this would remove any cost savings.
* Similarly, I think the paper overstates its relationship with coreset/data selection methods a little bit. These are generally not for speeding up an initial training run, but subsequent ones e.g. for continual learning or hyperparameter tuning and typically incur an upfront cost. On the contrary, the proposed method speeds up a given training run without producing any artefacts (a coreset) that are useful downstream. So to me this seems more like a curriculum learning paper (although I am not particularly familiar with this branch of the literature, so this is a somewhat speculative statement).

**Questions:**

* I would like to see results for Random*, $\epsilon$-greedy and UCB with annealing and gradient re-scaling (for Random*; for the other two as applicable). As much as possible for Table 1 and ideally Table 2 (ResNet-18 instead of 50 is perfectly fine if that makes it more realistic). My hypothesis here would be that all baselines will match InfoBatch in discriminative performance, which would necessitate the main claims in the paper being updated (again, I don't think this affects the value of the contribution). If all the baselines are already using annealing and rescaling, this point is of course void.
* Add a full data baseline with reduced epochs as proposed in the weaknesses.
* Is there anything more to section 2.3 than showing unbiasedness?
* I would be curious what fraction of the data are soft-pruned throughout training? With the losses being bound below by 0, I would expect the distribution to become quite asymmetric as training proceeds. Could e.g. the median or some percentile be preferable? The median (not sure about arbitrary percentiles) can be computed in linear time, although I don't know if it is possible to update it online as for the mean.
* Do you have any thoughts on how to set $r$ and $\delta$ on a new dataset/architecture? The benchmarks in the paper are of course well-studied, but I think it would be a nice addition to the paper to discuss some heuristics for finding close-to-optimal values without needing a full training run (although suboptimal values already present a cost saving of course).

Typos:
- p1, §3: "constrainT computation resources" -> "constrainED computation resources"
- p1, §4: "cubersome" -> "cumbersome"
- Section 3.3, §1: "see appendix" -> missing specific reference
- B4. last §: "bond" -> "bound"

---

> ### Author Response · Authors · 2023-11-20
> **Response to reviewer 8u8Q (1/4)**
>
> We sincerely thank the review 8u8Q for the meticulous review and responsible attitude.
> We fix those typos in the revised PDF, thank you.
>
> For the concerns and questions, here are our responses:
>
> **Q1: Are rescaling and annealing all the key points of InfoBatch?**
>
> **A1:** Thanks for the question. This is partially right. **Only with soft (probabilistic) pruning, recaling can take effect
> with no bias.** Rescaling for hard pruning would harm the performance. Annealing can be used in all settings, which helps to
> achieve lossless performance when rescaling is unbiased. To verify this, we conduct the following experiment on CIFAR-100 ResNet-50.
>
> | method                     | saving ratio  (%)| performance (%) |
> |----------------------------|------------|-------------|
> | InfoBatch w/o soft pruning | 33         | 80.1        |
> | InfoBatch w/ soft pruning | 33         | 80.6        |
>
> Note: training on the whole dataset achieves 80.6% acc.
>
> **Conclusion: Soft pruning is important to unbiased rescaling and achieving lossless results.**
>
> This question is not fully answered yet at this point. The further insight of InfoBatch's sample selection is discussed
> in answer A2 and A3.
>
> **Q2: (Generalization) Results for other dynamic baselines with annealing and rescaling**
> **A2:** Thanks for the question. The random in Table 5 (CIFAR-100 ResNet-50) is the random* with annealing and rescaling.
> Its result is improved as expected over table 4, since random* is also a kind of soft pruning.
>
>
> However on UCB which uses (sort and) hard pruning, the thing is different. The rescaling can only be applied to all remaining
> samples which are all higher score samples, being statistically different from pruned samples.
> We show the corresponding experimental results here:
>
> Setting: Train ResNet-50 on CIFAR-100 using UCB and random*. We report their acc and pruning ratio here.
>
> | method                                      | saving ratio (%) | performance  (%) | compared to InfoBatch |
> |---------------------------------------------|--------------|--------------|-----------------------|
> | UCB                                         | 33%          | 79.9         | -0.7                  |
> | UCB                                         | 41%          | 79.5         | -1.1                  |
> | UCB+anneal                                  | 33%          | 80.1         | -0.5                  |
> | UCB+anneal                                  | 41%          | 79.6         | -1.0                  |
> | UCB+rescale+anneal                          | 33%          | 79.9         | -0.7                  |
> | UCB+rescale+anneal                          | 41%          | 79.5         | -1.1                  |
> | random*                                     | 33%          | 79.7         | -0.9                  |
> | random*+rescale+anneal<br/>(Table 5 random) | 33%          | 80.5         | -0.1                  |
> | random*+rescale+anneal                      | 41%          | 80.3         | -0.3                  |
>
>
> **Analysis:**
> * UCB's performance is increased by annealing using the same saving ratio (by controlling the pruning fraction), but much lower than
> InfoBatch
> * Rescaling with annealing improves the performance of random* using the same saving ratio; but adding rescaling leads to degraded performance
> of UCB
>
> **Conclusion:** **Annealing could help to improve performance on a broader range of
> sample selection methods, but rescaling can only take effect if it is using soft pruning.**
>
> **Q3: Point beyond soft pruning, rescaling, and annealing**
>
> **A3:** Thanks for the comment.
>
> **It is possible to use different (static or dynamic) thresholds with different pruning ratios**
>
> * According to the proof in Section 2.3, the expectation rescale equation would hold independent of the threshold selection and probability.
>
> **It is possible to use a more aggressive ratio r on lower score (gradient) samples.**
>
> * According to Appendix B.3 (B.4 in revised version), $\mathop{\mathbb{E}}\_{rescaled}[G\_{z}^2/(1-\mathcal{P}\_t(z))]\leq \mathop{\mathbb{E}}\_{\mathcal{D}}[G_z^2]$
> could potentially lead to health rescaling.
> * We use the percentile method to prune part of the data (20%) with a more aggressive ratio r (0.75) as the default setting as claimed in 3.1.
> This leads to a higher pruning ratio without performance loss.
> In Table 6 with this improvement, CIFAR-100 ResNet-50 can achieve 41.3% saving ratio, compared to 33% in Table 5 using uniform r.
>
> **Summarization:**
> * Rescaling (w/ soft pruning) and annealing are our contributions which do generalize to difference threshold selection
> * We further explore better data efficiency with better utilization of scores from both theoretical and experimental aspects.

---

> ### Author Response · Authors · 2023-11-20
> **Response to reviewer 8u8Q (2/4)**
>
> **Q4: Add a full data baseline with reduced epochs and adjusted learning rate.**
> **A4:** Thanks for the question. Here we add the tuned learning rate baseline with reduced epoch numbers.
>
>
> | settings with adjusted learning rate         | saving ratio      | performance (%) | InfoBatch performance |
> |----------------------------------------------|-------------------|-----------------|-----------------------|
> | CIFAR-10 ResNet-18 baseline     | 30%   (140 epoch) | 94.8            | 95.6 ⭡ 0.8            |
> | CIFAR-10 ResNet-18 baseline     | 50%   (100 epoch) | 94.6            | 95.1 ⭡ 0.5            |
> | CIFAR-10 ResNet-18 baseline     | 70%   (60 epoch)  | 92.7            | 94.7 ⭡ 2.0            |
> | CIFAR-100 ResNet-18 baseline    | 30%   (140 epoch) | 77.0            | 78.2 ⭡ 1.2            |
> | CIFAR-100 ResNet-18 baseline    | 50%   (100 epoch) | 76.9            | 78.1 ⭡ 1.2            |
> | CIFAR-100 ResNet-18 baseline    | 70%   (60 epoch)  | 76.3            | 76.5 ⭡ 0.2            |
> | ImageNet-1K ResNet-50 baseline  | 60%   (54 epoch)  | 72.8            | 76.5 ⭡ 3.7            |
>
> **Analysis**
> * InfoBatch outperforms the tuned baseline. This is properly because higher loss samples are
> more sensitive to learning rate scaling as discussed in Appendix B.3.
> * Based on that, when scaling the learning rate to match
> the original training progress, scaling lower loss samples is still preferred than higher loss samples.
>
> [//]: # (We add this baseline to Table 1 in the revision.)
>
> **Conclusions:**
> * With the same computation, InfoBatch has higher performance than the tuned baseline.
> * According to the table and the point above, the improvement of InfoBatch is not caused by a lack of tuning of the benchmark.
>
>
> **Q5: Purpose of 2.3**
>
> **A5:** Thanks for the question. Previously, considering the different backgrounds of reviewers, we wrote 2.3 with more detailed
> steps. We summarize it to make it brief (in the updated revision), the changes are as follows:
>
> * We merge previous Eqn. (6) and (7) to revised Eqn. (6), and move the middle step to B.1.
> * We merge previous Eqn. (8) and (9) to revised Eqn. (7).
>
> **Q6: "Info" and "lossless" in the title**
>
> **A6:** Thanks for the comment.
> * We call our method "InfoBatch" because it takes advantage of the current learning status (loss) to do selective data pruning. It is "informative"
> instead of purely random.
> * We achieve lossless results on many tasks with substantial savings, which is an important milestone. It is true InfoBatch cannot
> guarantee a lossless result in all settings, but the empirical hyperparameters have already achieve lossless results on image classification,
> semantic segmentation, image generation, and language model finetuning.
>
> We are welcome to further discussion on this problem during the author reviewer discussion period.
>
>
> **Q7: What fraction of the data are soft-pruned throughout training?**
>
> **A7:** Thanks for the question. We plot the saving fraction curve of InfoBatch in CIFAR-10 ResNet-50 training in
> revised Appendix E.1 Fig.6.
>
> For **entropy loss**, InfoBatch (mean) usually **starts with a relatively lower ratio,
> and then the ratio keeps increasing till the end**. This is probably because maximizing log probability tends to converge to a
> long-tailed loss distribution.
>
> We show the loss distribution visualization during training in revised Appendix E.1 Fig.7. We can see the loss distribution
> is initially right-skewed and finally left-skewed.

---

> > ### Author Response · Authors · 2023-11-20
> > **Response to reviewer 8u8Q (3/4)**
> >
> > **Q8: Could the median or some percentile be preferable for threshold selection?**
> > **A8:** Thanks for the question. Some percentile could be preferable given enough effort of tuning.
> >
> > In 3.1 experiment details, we briefly mentioned we use a more aggressive pruning probability r(0.75)
> > for smaller loss samples (lowest 20%) on CIFAR-100, ImageNet-1K, and ADE20K experiments.
> > Here we elaborate on more details of these explorations.
> >
> > The percentage thresholding is implemented with the numpy percentile method taking a linear time (still amortized O(1)).
> > We did some ablation on CIFAR-100 ResNet-50 as follows:
> >
> > **Setting:** InfoBatch on CIFAR-100 ResNet-50 with different thresholding method. A percentage (p%) corresponds to a threshold
> > where that much (p%) of data is smaller than the threshold.
> > All results are averaged across three runs with std reported.
> > We report their acc and pruning ratio here.
> >
> > | range:ratio(r)                                                         | saving (%) | Acc (%)      |
> > |------------------------------------------------------------------------|-----------|--------------|
> > | full data (r=0)                                                        | 0         | **80.6±0.1** |
> > | <mean:0.5                                                              | 33        | **80.6±0.1** |
> > | [0%,76%):0.5                                                           | 33        | **80.6±0.1** |
> > | [0%,85%):0.5                                                           | 36.8      | 80.4±0.1     |
> > | [0%,20%):0.75,[20%,85%):0.5                                            | 41.3      | **80.6±0.2** |
> > | [0%,30%):0.75,[30%,90%):0.5                                            | 45        | 80.4±0.2     |
> > | [0%,36%):0.75,[36%,95%):0.5                                            | 49        | 80.3±0.2     |
> > | [0%,20%):0.75，[20%,40%):0.7，[40%,60%):0.65，[60%,80%):0.6，[80%,90%):0.5 | 51.3      | 80.1±0.2     |
> >
> >
> > **Comparison:**
> > 1. The percentile method uses a fixed pruning ratio for each epoch, while the mean threshold is adaptive. The two thresholding methods can both achieve reasonable performance.
> > 2. Mean thresholding generally doesn't need much tuning.
> > The percentile method may cost more tuning but potentially a higher saving ratio, especially when using different pruning probabilities for different score ranges.
> >
> > **Conclusion: if the tuning budget is limited, mean thresholding could be preferred; if the tuning budget is sufficient, quantile methods
> > could get a higher saving ratio using different pruning probabilities for different score ranges.**
> >
> > **Q9: Relation to curriculum learning**
> > **A9:** Thanks for the comment. Our work shares some similarities with curriculum learning while differing in many aspects.
> > We summarize it as follows:
> >
> > | feature\method                            | InfoBatch                               | Curriculum Learning ([1] as an example)                     |
> > |-------------------------------------------|-----------------------------------------|-------------------------------------------------------------|
> > | Calculation before training               | No                                      | Yes                                                         |
> > | Change sample frequency                   | Yes                                     | Yes                                                         |
> > | Arrange method                            | Statistical                             | Sort                                                        |
> > | Scoring Overhead                          | O(1) per sample, obtained from training | At least one forward pass per sample (O(M)) before training |
> > | Overall Cost                              | Negligible                              | Non-negligible                                              |
> > | Purpose                                   | Acceleration                            | Performance Improvement                                     |
> > | Adaptive to learning status               | Yes                                     | No                                                          |
> > | Pruned Part                               | Easy                                    | Hard (Standard)                                             |
> > | CIFAR-100 ResNet-50 Saving at Convergence | 40%                                     | 10%                                                         |
> >
> > [1] Wu, Xiaoxia, Ethan Dyer, and Behnam Neyshabur. "When do curricula work?." arXiv preprint arXiv:2012.03107 (2020).

---

> ### Author Response · Authors · 2023-11-20
> **Response to reviewer 8u8Q (4/4)**
>
> **Q10: How to set r and $\delta$ on a new dataset/architecture?**
>
> **A10:** Thanks for the good question.
>
> We consider the r value in the future work (see it in Appendix B.4), we plan to use
> $\mathbb{E}\_\text{rescaled}[G\_{z}^2/(1-\mathcal{P}\_t(z))]\leq {\mathbb{E}}\_{\mathcal{D}}[G_z^2]$
> to find near-optimal r. We theoretically analyze the reasonableness of choosing r according to this equation.
> * Our proposed procedure is to measure the $\mathop{\mathbb{E}}_{\mathcal{D}}[G_z^2]$
> (or some cheaper indicator values instead) after warmup, then use
> the statistics to quickly select near optimal r values.
>
> For delta, our default value 0.875 is actually corresponding to the annealing stage of the one cycle scheduler
> we used for the learning rate schedule. It corresponds to a drop in loss during normal training.
> In general, it is the last 17.85% of a cosine annealing. As cosine annealing is prominent in
> many tasks, we suggest **last 0.1785 fraction of cosine annealing could be used as default**.

---

> > ### Author Response · Authors · 2023-11-21
> > **Looking forward to the reply**
> >
> > Dear reviewer 8u8Q:
> >
> > Thanks so much again for the time and effort in our work.
> > According to the comments and concerns, we conduct the corresponding experiments and further discuss the related points.
> > Additionally, we have revised our writing of 2.3 and provided visualization plots in Appendix E.1.
> >
> > As the rebuttal period is about to close, may I know if our rebuttal addresses the concerns? If there are further concerns or questions, we are welcome to address them.
> > Thanks again for taking the time to review our work and provide insightful comments.

---

> > > ### Comment · Reviewer_8u8Q · 2023-11-21
> > >
> > > Thank you for the extensive additional results. These largely address my concerns and I will increase my score.
> > >
> > > I'm happy to believe that the soft pruning and rescaling/annealing techniques in the paper indeed lead to a meaningful saving in compute costs (and that this is not just due to suboptimal default hyperparameters for benchmarks). The one aspect that I remain a little bit sceptical on is that of the decision criterion. I appreciate that this is based on a fairly limited number of results, but it does look to me like random*+rescale+anneal does essentially as well as pruning based on loss values. I think it would be insightful for the camera-ready paper to test this baseline in a couple more settings and see if the observation holds across the board (in which case section 2.3 should be updated) or if e.g. for larger savings ratios we see the informative pruning criterion performing better.

---

> ### Author Response · Authors · 2023-11-21
>
> Dear Reviewer 8u8Q,
>
> Thanks for acknowledging our work. We agree with the advice, and we are running the corresponding experiments to report the
> observations (how much does loss-guided pruning differ from random pruning in performance under our framework) as follows:
> 1. With **various architectures**, we plan to evaluate the performance on CIFAR-10 with random/loss-guided pruning + rescaling + annealing at default saving ratio
>
> | architecture\selection | Random | Loss Guided |
> |------------------------|--------|-------------|
> | VGG                    | *      | *           |
> | ResNet-18              |95.3     | 95.6         |
> | ResNet-50              | 95.3      | 95.6          |
> | Swin-Tiny               | *      | *           |
>
> **Note:** * denotes running experiment waiting for the result. We will keep updating the results till the end of the reviewer-author discussion period. We will discuss these results in the revision.
>
> 2. On **various datasets**, we plan to evaluate the performance with ResNet-50 at default saving ratio
>
> | dataset\selection | Random | Loss Guided |
> |-------------------|--------|-------------|
> | CIFAR-10          | 95.3      | 95.6          |
> | CIFAR-100         | 80.5  $\pm$ 0.2  | 80.6 $\pm$ 0.1          |
> | ImageNet-1K       | *      |  76.5        |
>
>
> **Note:** * denotes running experiment waiting for the result. We will keep updating the results till the end of the reviewer-author discussion period. We will discuss these results in the revision.
>
> 3. For **higher saving ratios**, we plan to evaluate the corresponding performance on CIFAR-10 ResNet-18 (using the aforementioned quantile method to control the ratio for loss-guided pruning)
>
> | ratio\selection | Random | Loss Guided |
> |-----------------|--------|-------------|
> | 0%(baseline)    | 95.6    | 95.6         |
> | 30%             | 95.3      | 95.6           |
> | 40%             | 95.3      | 95.5           |
> | 50%             | 94.9      | 95.2           |
> | 60%             | 94.8      | 95.1           |
> | 70%             | *      | *           |
>
>
> **Note:** * denotes running experiment waiting for the result. We will keep updating the results till the end of the reviewer-author discussion period. We will discuss these results in the revision.
>
> Once again, we appreciate the insightful and constructive comments on our work. We will be glad to improve our work if
> there is any following advice. Thanks again for your comments and time.

---

> > ### Public Comment · ~Ziheng_Qin1 · 2024-05-10
> > **Update the remaining results**
> >
> > 1. With **various architectures**, we plan to evaluate the performance on CIFAR-10 with random/loss-guided pruning + rescaling + annealing at default saving ratio
> >
> > | architecture\selection | Random | Loss Guided |
> > |------------------------|--------|-------------|
> > | VGG                    | 93.5   | 93.9        |
> > | ResNet-18              | 	95.3  | 	95.6       |
> > | ResNet-50              | 	95.3  | 	95.6       |
> > | Swin-Tiny               | 80.95  | 85.03       |
> >
> > 2. On **various datasets**, we plan to evaluate the performance with ResNet-50 at default saving ratio
> >
> > | dataset\selection | Random          | Loss Guided |
> > |-------------------|-----------------|-------------|
> > | CIFAR-10          | 95.3            | 95.6          |
> > | CIFAR-100         | 80.5  ± 0.2 | 80.6 ± 0.1          |
> > | ImageNet-1K       | 76.5            |  76.5        |
> >
> > 3. For **higher saving ratios**, we plan to evaluate the corresponding performance on CIFAR-10 ResNet-18 (using the aforementioned quantile method to control the ratio for loss-guided pruning)
> >
> > | ratio\selection | Random | Loss Guided |
> > |-----------------|--------|-------------|
> > | 70%             | 94.7   | 94.7        |
> >
> > The results suggest that loss-guided sample selection is preferred over random selection in more cases (e.g. VGG, swin-tiny, r18/r50). It aligns with the theoretical analysis in Appendix B, thus the result didn't change our main conclusion and method.

---

### Official Review · Reviewer_nCAL · 2023-11-01

**Soundness:** 2 fair
**Presentation:** 4 excellent
**Contribution:** 3 good
**Rating:** 6
**Confidence:** 4

**Summary:**

This paper presents InfoBatch, a novel data pruning approach which dynamically determines pruning probability over the course of training. InfoBatch soft prunes data with small loss value leading to negligible training cost overhead, and rescales remaining data so as to achieve unbiased gradient expectation. By conducting experiments across a wide range of tasks, the paper demonstrates the effectiveness and robustness of InfoBatch as a state-of-the-art data pruning technique in terms of tradeoff between performance and computational cost.

**Strengths:**

- The paper tackles a practically-relevant problem supported by a fair amount of experiments conducted across various tasks in the image domain.
- InfoBatch is simple yet has a distinctive benefit over existing dynamic data pruning approaches: (i) replacing sorting operation with mean-thresholding significantly reduces the overhead cost, and (ii) gradient expectation bias is well addressed supported by theoretical analysis.
- The proposed method demonstrates superior performance compared to the baselines, and the paper provides a comprehensive review of relevant previous works.
- Overall, the paper is well-written and easy to follow.

**Weaknesses:**

- InfoBatch improves over UCB via throwing away the dataset sorting operation. However, in Table 2, the practical overhead cost saving seems negligible compared to the wall clock time and the total node hour. Also, how was the overall saved cost calculated in Tables 6 and 7?
- To my understanding, annealing utilizes the whole dataset without pruning for the last 0.125% of total training epochs. This raises several concerns: (i) As the wall clock time of UCB and InfoBatch in Table 2 are both 10.5h, is this value taking the annealing process into account? (ii) Regarding Table 1, all the baselines and InfoBatch are compared under the same dataset pruning ratio. I wonder whether this is a fair comparison when annealing is involved in InfoBatch. (iii) Why did the authors utilize annealing only as a means of stabilizing the optimization, rather than leveraging the full dataset at the very beginning of optimization when we know that the early epochs of training can heavily influence the convergence of the loss landscape [1]?
- The authors may need to provide further clarification regarding how annealing contributes to the stabilization of the rescaling process, especially if it does not seem to significantly impact the variance of the results in Table 4.
- The authors claim that the use of loss values in pruning conditions serves two purposes: (i) it reflects the learning status of samples, and (ii) it theoretically ensures unbiased gradient expectations. However, in Table 5, it is observed that even a random pruning criterion achieves nearly the same performance as the original pruning condition. This result raises questions about the necessity and effectiveness of using loss values as a pruning criterion and may require further discussion or clarification in the paper.

**Questions:**

- How many random seeds are used throughout the experiments?


[1] Fort et al., “Deep learning versus kernel learning: an empirical study of loss landscape geometry and the time evolution of the Neural Tangent Kernel.” 2020.

---

> ### Author Response · Authors · 2023-11-20
> **Response to reviewer nCAL (1/2)**
>
> We sincerely thank the reviewer nCAL for the valuable questions and comments.
> For the concerns and questions, here are our responses:
>
> **Q1: Overhead cost saving over UCB seems negligible compared to training**
>
> **A1:** Thanks for the question. Sort is O(NlogN) for full data and amortized O(logN) for each sample, ours is O(N) and
> amortized O(1). We design the operation considering it could differ by a large factor when N is very big.
>
> This overhead saving of operation compared to UCB is 10 times (0.03h to 0.0028h) on ImageNet-1K; however we agree it is
> not a significant improvement compared to training time saving (17.5h->10h). Our main saving comes from saving the forward and backward
> computation, which is shown in the table below:
>
>
> **Setting:** ResNet-50 on CIFAR-100. All results are averaged across three runs with std reported.
> We report their acc and pruning ratio here.
>
> |           | Overhead (s) | Saving      | Acc. (%) |
> |-----------|--------------|-------------|----------|
> | UCB       | < 2          | 20%, ~800s  | 80.4±0.2 |
> | UCB       | < 2          | 33%, ~1320s | 79.8±0.2 |
> | UCB       | < 2          | 41%, ~1640s | 79.5±0.2 |
> | InfoBatch | < 0.1        | 33%, ~1320s | 80.6±0.1 |
> | InfoBatch | < 0.1        | 41%, ~1640s | 80.6±0.2 |
>
> **Analysis**:
> * At lossless performance, InfoBatch can save 41% compared to 20% of UCB. The saved time is a 100% improvement.
> * At the same saving ratio, UCB has a much higher performance degradation
>
> We degraded the claim of the part on time complexity, which can be visible in the new revision. The main modifications can
> be summarized as:
> 1. in section 4 related works, we change "which could be a non-negligible overhead" -> "which could be an overhead"
>
>
>
> **Q2: How was the overall saved cost calculated in Tables 6 and 7**
>
> **A2:**
> Thanks for the question. As the overhead is negligible compared to training, we measured wall clock time and found the sample iteration saving
> is basically the same as wall clock saving. So the **overall saved cost is both wall clock time and sample iteration**.
>
> **Q3: Is annealing taken into account for saving?**
>
> **A3:** Thanks for the question. The answer is yes. Our time = overhead + pruned training time + annealing training time.
> For table 1, InfoBatch saves slightly more cost than reported, with annealing already taken into account.
>
> **Q4: Using full dataset at warmup**
>
> **A4:** Thanks for the question. Only in the experiment of Timm Swin-Tiny training, we warmup with the full dataset for 5 epochs to avoid training instability.
>
> On other tasks,the initial pruning ratio is lower, which serves as a pruning warmup for CNN-based experiments. A fraction
> is visualized in revised Appendix E.1 Fig.6. It is properly due to early stage loss distribution is right-skewed and later left-skewed,
> which is visualized in revised Appendix E.1 Fig.7.
> ViT-based networks seem more sensitive to this warmup stage than CNN-based ones.
>
> Thank you for pointing it out, we make the following modification to make it more clear:
> * In sec A.3 paragraph 2, add "We also warmup InfoBatch for 5 epochs, only recording scores without pruning."

---

> ### Author Response · Authors · 2023-11-20
> **Response to review nCAL (2/2)**
>
> **Q5: Further clarification of how annealing contributes to stabilization**
>
> **A5:** Thanks for the question. We found Table 4 with a low number of precision digits is hard to illustrate this point clearly.
> We draw a plot in the revised E.2 Fig.8. All experiments
> use the same saving ratio.
>
> **Analysis:** In this plot, compared to only rescaling, adding annealing leads to an increased mean performance and slightly reduced performance variance.
>
> **Conclusion:** This plot indicates how annealing helps stabilize the training.
>
> **Q6: Necessity of using loss value**
> **A6:** Thanks for the comment.
>
> By reasonably utilizing loss, we can further improve data efficiency, based on the inference in Appendix B.3 (revised version B.4).
>
> Noted that in Table 5 (CIFAR-100 R-50), the lossless pruning ratio using mean is 33%. In Section 3.1 experiment details
> we claim to use a more aggressive pruning probability r(0.75) for smaller loss samples (lowest 20%) on CIFAR-100;
> that leads to a 41.3% saving ratio in Table 6.
>
> We conduct the following experiment to elaborate:
>
> **Setting:** InfoBatch on CIFAR-100 ResNet-50 with increased r for certain samples.
> All results are averaged across three runs with std reported.
> We report their acc and pruning ratio here.
>
> | method                     | saving ratio (%) | performance (%) | time    |
> |----------------------------|------------------|-----------------|---------|
> | lower loss higher r(0.75)  | 41.3             | 80.6±0.2        | 1.48h   |
> | higher loss higher r(0.75) | 41.3             | 79.9±0.3        | 1.48h   |
> | random                     | 41.3             | 80.3±0.2        | 1.48h   |
>
> **Analysis:**
> * Using a higher pruning ratio r for higher loss samples significantly degrades performance.
> * A related discussion is in Appendix B.3 (revised version B.4) about why lower-loss samples are preferred.
> $\mathbb{E}\_\text{rescaled}[G\_{z}^2/(1-\mathcal{P}\_t(z))]\leq {\mathbb{E}}\_{\mathcal{D}}[G_z^2]$ could be
> the condition for a healthy rescaling.
> * Compared to random selection, there is almost no overhead to use loss, as it can be obtained with no extra cost.
>
> **Conclusions:**
> * **Loss is preferred, especially when using an adaptive high pruning ratio** which further improves data
> efficiency.
> * The **overhead of loss is negligible**, so its **improvement is of almost no cost** and thus also preferred in settings
> without adaptive high pruning ratio.
>
>
> **Q7: How many random seeds are used throughout the experiments**
>
> A7: Thanks for the question. For CIFAR-10/100 experiments, we measured at least 3 trials for those values with std.

---

> ### Author Response · Authors · 2023-11-21
> **Looking forward to the reply**
>
> Dear reviewer nCAL:
>
> Thanks so much again for the time and effort in our work.
> According to the comments and concerns, we conduct the corresponding experiments and further discuss the related points.
> Besides, we have revised our claim of overhead compared to dynamic methods. We also provided visualization plots in Appendix E.1 to further illustrate how annealing contributes to stabilization.
>
> As the discussion period is nearing its end, please feel free to let us know if there are any other concerns. Thanks again for your time and efforts.

---

> > ### Author Response · Authors · 2023-11-22
> >
> > Dear Reviewer nCAL,
> >
> > Considering the limited time available, and in order to save the reviewer's time, we summarized our responses here.
> >
> > **1. [Improvement compared to UCB]**
> >
> > Question: Overhead cost saving seems negligible compared to the wall clock time and the total node hour.
> >
> > Response:
> > The major improvement is in the lossless saving ratio, as on CIFAR-100 ResNet-50, InfoBatch saved 41% compared to 20% of UCB.
> > The saved time is a 100% improvement (800s->1640s). We agree in Table 2 the overhead saving (0.03h to 0.0028h) is not
> > a significant improvement compared to training time saving (17.5h->10h). We degraded the corresponding claim of the part about time complexity in the updated revision.
> >
> > **2. [Details explanation]**
> >
> > Question:
> > (a) How was the overall saved cost calculated in Tables 6 and 7?
> > (b) Is annealing taken into account for saving?
> > (c) How many random seeds used
> >
> > Response:
> > (a) As the overhead is negligible compared to training, the overall saved cost is both wall clock time and sample iteration.
> > (b) Our time = overhead + pruned training time + annealing training time. For table 1, InfoBatch saves slightly more
> > cost than reported, with annealing already taken into account (not contributing to saving).
> > (c) For CIFAR-10/100 experiments, we measured at least 3 trials for those values with std.
> >
> > **3. [Full data at warmup]**
> >
> > Question:
> > Using full dataset at warmup.
> >
> > Response:
> > In the experiment of Timm Swin-Tiny training, we warm up with the full dataset for 5 epochs to avoid training instability.
> > On other tasks, the mean has an adaptive pruning ratio which probably serves as a pruning warmup (revised Appendix E.1 Fig 6).
> > ViT-based networks seem more sensitive to this warmup stage than CNN-based ones.
> >
> > **4. [Annelaing's effect]**
> >
> >
> > Question:
> > May need to provide further clarification regarding how annealing contributes to the stabilization of the rescaling process.
> >
> > Response:
> > We draw a plot in the revised E.2 Fig.8. All experiments use the same saving ratio. In this plot, compared to only rescaling,
> > adding annealing leads to an increased mean performance and slightly reduced performance variance.
> >
> > **5. [Clarification of necessity and effectiveness of using loss values as a pruning criterion]**
> >
> > Question:
> > Further clarify the necessity and effectiveness of using loss values as a pruning criterion.
> >
> > Response:
> > **By reasonably utilizing loss, we can further improve data efficiency,
> > based on the inference in Appendix B.3 (revised version B.4)**. In table 5 (CIFAR-100 R-50), the lossless pruning ratio using the mean is 33%.
> > In section 3.1 experiment details we claim to **use a more aggressive pruning probability** r(0.75)
> > **for smaller loss samples** (lowest 20%) on CIFAR-100; that leads to a 41.3% saving ratio in Table 6. We demonstrate the additional experiment showing **Pruning high-loss samples with an aggressive ratio would harm the performance significantly**.
> > **Loss is preferred over random, especially when using an adaptive high pruning ratio; The overhead of loss is negligible, thus its improvement is of almost no cost.**
> >
> >
> > Since the discussion period is about to close soon, could we know if our responses addressed your concerns? Please feel free to let us know if there are any other concerns. Thanks!

---

> > ### Comment · Reviewer_nCAL · 2023-11-22
> >
> > I appreciate the authors for their detailed response and the supplementary experiments. As the authors addressed most of my concerns, including clarification of total saving overhead cost and warmup stage with full dataset, I will raise my score from 5 to 6, and vote for the acceptance.
> >
> > - I have one more lingering question regarding the stabilization effect of annealing. While I agree with the most of the authors' response, I still remain uncertain whether the primary role of annealing is to reduce variance. In Fig. 8, it appears that annealing notably improves performance, yet it doesn't seem to stabilize performance as effectively as when it is used solely. I would appreciate if the authors could clarify why they mainly portray annealing as a stabilizing factor in Section 3.4?

---

> > > ### Author Response · Authors · 2023-11-22
> > >
> > > Dear reviewer nCAL,
> > >
> > > Thanks for appreciating our work and giving the valuable comment. It is a point worth discussing.
> > >
> > > **Previous considerations:** In experiments, we found that without annealing (only rescaling), InfoBatch may still
> > > achieve lossless results with a lower probability, so we interpreted it as annealing stabilized the performance.
> > >
> > > **Reflection:** In the claim of 2.4, the primary role of annealing is to reduce the _remaining gradient bias_. The remaining gradient bias would
> > > cause the performance mean degradation and variance increase. According to the feedback, we found "reduce performance variance"
> > > in 3.4 ablation experiment of $\delta$ is not comprehensive enough to represent all the roles of annealing.
> > >
> > > **Modification:** We made the following changes in 3.4 ablation experiment of $\delta$ in the updated revision:
> > > * Page 7 Sec 3.4 last paragraph: "To further reduce the performance variance" -> "To reduce the remaining gradient expectation bias"
> > > * Page 8 Sec 3.4 following the previous page: "leaving more performance variance" -> " leaving more remaining gradient bias"
> > > * Page 8 Sec 3.4 following the previous page: "which may result in degraded average performance" -> "which may result in degraded average performance and increased performance variance"
> > >
> > > Thanks again for your time and comments. We will be glad to address other concerns if there are any.

---

> > > > ### Comment · Reviewer_nCAL · 2023-11-22
> > > >
> > > > Thank you for the further clarification. With my concerns fully addressed, I vote in favor of acceptance.

---

### Official Review · Reviewer_Nce2 · 2023-11-01

**Soundness:** 4 excellent
**Presentation:** 4 excellent
**Contribution:** 3 good
**Rating:** 8
**Confidence:** 4

**Summary:**

This work proposes a dynamic data pruning approach that can obtain lossless performances with less training cost.
It achieved the unbiased gradient update by randomly pruning a portion of less informative samples and rescaling the gradient of the remaining samples. The proposed approach consistently obtains lossless training results on various ML tasks.

**Strengths:**

- Clear presentation and easy-to-follow writing.
- The proposed method is theoretically-supported and, more importantly, very efficient and easy to implement.
- The evaluation, together with the analysis, is extensive and convincing.

**Weaknesses:**

The paper conducts a complete study on dynamic data pruning, and the following weakness is relatively minor.
- Missing recent works: 1) static data pruning [a,b,c], 2) dynamic data pruning [d]

---
[a] Active learning is a strong baseline for data subset selection. NeurIPS workshop, 2022

[b] Moderate: Moderate coreset: A universal method of data selection for real-world data-efficient deep learning. ICLR, 2023

[c] CCS: Coverage-centric Coreset Selection for High Pruning Rates. ICLR, 2023

[d] Prioritized Training on Points that are Learnable, Worth Learning, and Not Yet Learnt. ICML, 2022

**Questions:**

- How to illustrate the gradient trajectory with landscape in Fig 1? Is it an illustration or a real plot on some dataset?
- Which dataset is used for Table 4? maybe ImageNet?

---

> ### Author Response · Authors · 2023-11-20
> **Response to reviewer Nce2**
>
> We sincerely thank the reviewer Nce2 for pointing out the missing references as well as a potential improvement. We make responses as follows.
>
> **Q1: Add missing references.**
>
> **A1:** Thanks for the advice. We update the section 1 and 4 to add those references(see the revision). The main changes are:
> * Sec 1, paragraph 2: add reference " Park et al., 2022; Xia et al., 2023; Zheng et al., 2023"
> * Sec 4, paragraph 1: add "AL (Park et al., 2022) propose
> to use active learning methods to select a coreset. Moderate (Xia et al., 2023) proposed to use the
> median of different scores as a less heuristic metric. Coverage-centric Coreset Selection (Zheng et al.,2023)
> additionally considers distribution coverage beyond sample importance. "
> * Sec 4, paragraph 2: add "Mindermann et al. (2022) propose Reducible Holdout Loss Selection which
> prioritizes samples neither too easy nor too hard. It emphasizes training learnable samples."
>
> **Q2: How to illustrate the gradient trajectory with landscape in Fig 1? Is it an illustration or a real plot on some dataset?**
>
> **A2:** Thanks for the question. Fig 1 is an illustration instead of a real plot. To avoid misunderstanding, we modify the caption
> "visualiztion" -> "illustration".
> The corresponding accuracy values are from Table 1 CIFAR100 experiment, comparing InfoBatch and EL2N-2 at 50% pruning ratio.
>
> **Improvement Plan:**
> We notice that there could be a potential improvement of this illustration using the method proposed in [1].
> It would express the same idea but with a more accurate visualization. We found in [2] Fig 1 they show the gradient sketch
> on such a loss landscape. We are welcome to further discussion on this in the auther reviewer discussion period (whether change our Fig 1 like that).
>
> [1] Li, Hao, et al. "Visualizing the loss landscape of neural nets." Advances in neural information processing systems 31 (2018).
> [2] Liu, Zhuang, et al. "Dropout Reduces Underfitting." arXiv preprint arXiv:2303.01500 (2023).
>
> **Q3: Which dataset is used for Table 4?**
>
> **A3:** Thanks for the comment. In section 3.4 first paragraph, we claim: _if not stated, the experiments are conducted on CIFAR-100 by default_.
> Table 4 is conducted on CIFAR100 as default dataset for ablation. We update its caption to make it more clear in the updated revision.

---

> > ### Comment · Reviewer_Nce2 · 2023-11-21
> > **Reviewer response**
> >
> > Great work! The authors addressed all my concerns and please include the real plot (as in [2] Fig 1) in the final version. Concerning InfoBatch's broad applicability, I believe there is a substantial impact. I will keep my score.

---

> > > ### Author Response · Authors · 2023-11-22
> > >
> > > Dear reviewer Nce2,
> > >
> > > We would like to express our sincere gratitude to reviewer Nce2 for acknowledging our work and
> > > providing constructive suggestions. We will update the Fig 1 accordingly in the revision. Thanks again for the time
> > > and effort in reviewing our work.

---

### Author Response · Authors · 2023-11-23
**Refined some minors**

Dear ACs and reviewers,

We sincerely thank you for the time and effort in our work. We went through our work and fixed some minors. We will continue to refine our work.

Thanks,

Authors of submission 247

---

### Meta-Review · Area_Chair_YRoq · 2023-12-08

**Metareview:**

All the reviewers are supportive of accepting this paper for ICLR. InfoBatch is a dynamic data pruning approach that adjusts the probability of pruning the data point throughout the training. The authors show results in a wide variety of settings including classification, detection, segmentation, and instruction finetuning. The paper is comprehensive and well-written. I would encourage the authors to address the concerns raised by the reviewers.

**Justification For Why Not Higher Score:**

N/A

**Justification For Why Not Lower Score:**

I feel this paper is well-written. Additionally, the authors carefully addressed all the issues pointed out by the reviewers and I'm happy with the paper overall, assuming the authors can address all concerns raised. Additionally, I believe this is an important problem and domain that needs principled solutions.

---

### Decision · Program_Chairs · 2024-01-16

Accept (oral)